# Trace gas oxidation sustains energy needs of a thermophilic archaeon at suboptimal temperatures

Pok Man Leung [1] ✉, Rhys Grinter [1], Eve Tudor-Matthew [1], James P. Lingford[1], Luis Jimenez[1], Han-Chung Lee[2], Michael Milton[1], Iresha Hanchapola[2], Erwin Tanuwidjaya[2], Ashleigh Kropp[1], Hanna A. Peach[3], Carlo R. Carere [3,4], Matthew B. Stott [3,5], Ralf B. Schittenhelm [2] & Chris Greening [1] ✉

Diverse aerobic bacteria use atmospheric hydrogen ($H_2$) and carbon monoxide (CO) as energy sources to support growth and survival. Such trace gas oxidation is recognised as a globally significant process that serves as the main sink in the biogeochemical $H_2$ cycle and sustains microbial biodiversity in oligotrophic ecosystems. However, it is unclear whether archaea can also use atmospheric $H_2$. Here we show that a thermoacidophilic archaeon, *Acidianus brierleyi* (Thermoproteota), constitutively consumes $H_2$ and CO to subatmospheric levels. Oxidation occurs across a wide range of temperatures (10 to 70 °C) and enhances ATP production during starvation-induced persistence under temperate conditions. The genome of *A. brierleyi* encodes a canonical CO dehydrogenase and four distinct [NiFe]-hydrogenases, which are differentially produced in response to electron donor and acceptor availability. Another archaeon, *Metallosphaera sedula*, can also oxidize atmospheric $H_2$. Our results suggest that trace gas oxidation is a common trait of Sulfolobales archaea and may play a role in their survival and niche expansion, including during dispersal through temperate environments.

Over the last 15 years, it has been established that aerobic bacteria residing in soils and other aerated ecosystems are the primary biogeochemical sink of the global hydrogen ($H_2$) cycle[1]. These bacteria consume atmospheric hydrogen gas (tropospheric concentration ~0.53 ppmv / ~ 0.4 nM)[2] using high-affinity [NiFe]-hydrogenases from the subgroups 1h, 1l, 1f, and 2a[3–7]. They relay $H_2$-derived electrons primarily to the aerobic respiratory chain, enabling ATP synthesis, though some bacteria also use them to sustain $CO_2$ fixation[1,5,8]. Diverse organoheterotrophic bacteria depend on atmospheric $H_2$ as an energy source to survive during organic carbon limitation, as confirmed by

genetic studies showing high-affinity hydrogenases enhance the long-term survival of *Streptomyces avermitilis* exospores[3,9] and *Mycobacterium smegmatis* cells[10–12]. Some bacteria also mixotrophically grow by co-oxidizing atmospheric $H_2$ with other energy sources, as inferred from studies on the organotroph *Sphingopyxis alaskensis*[13], methanotroph *Methylocapsa gorgona*[14], nitrite oxidizer *Nitrospira moscoviensis*[15], and the iron and sulfur oxidizer *Acidithiobacillus ferrooxidans*[16]. In temperate soils, genome-resolved metagenomic analysis suggests 32% of bacterial cells and at least 21 phyla encode hydrogenases to oxidize atmospheric $H_2$[17,18]. Moreover, atmospheric

[1]Department of Microbiology, Biomedicine Discovery Institute, Monash University, Clayton, VIC 3800, Australia. [2]Monash Proteomics and Metabolomics Platform and Department of Biochemistry, Monash Biomedicine Discovery Institute, Monash University, Clayton, VIC 3800, Australia. [3]Geomicrobiology Research Group, Department of Geothermal Sciences, Te Pū Ao | GNS Science, Wairakei, Taupō 3377, Aotearoa New Zealand. [4]Te Tari Pūhanga Tukanga Matū | Department of Chemical and Process Engineering, Te Whare Wānanga o Waitaha | University of Canterbury, Christchurch 8140, Aotearoa New Zealand. [5]Te Kura Pūtaiao Koiora | School of Biological Sciences, Te Whare Wānanga o Waitaha | University of Canterbury, Christchurch 8140, Aotearoa New Zealand. ✉ e-mail: bob.leung@monash.edu; chris.greening@monash.edu

$H_2$ oxidation sustains the energy, carbon, and hydration needs of bacteria in ecosystems with low primary production, such as Antarctic desert soils[7,19]. Atmospheric carbon monoxide (CO; tropospheric concentration ~0.09 ppmv / 0.086 nM)[20] is also a vital energy source sustaining the survival of aerobic bacteria[21,22]. Similar to atmospheric $H_2$, this gas is oxidized by high-affinity form I CO dehydrogenases and the derived electrons are transferred to terminal oxidases[21–23]. Overall, atmospheric $H_2$ and CO are dependable lifelines for bacterial survival given their ubiquity, diffusibility, and high-energy content[24].

To date, atmospheric $H_2$ oxidation has not been observed in the domain archaea. Aerobic archaea are abundant community members in oxic environments, accounting for 1–5% in surface soils[25,26] and 2–20% in ocean waters[27]. In extreme environments such as oxic sub-seafloor sediments, acid mine drainages, salt lakes, and hot springs, aerobic archaea are relatively enriched and can constitute over half the microbial population[28,29]. Following the seminal report of aerobic $H_2$ oxidation by thermoacidophilic archaea 30 years ago by Stetter et al.[30], numerous other isolates from geothermal habitats associated exclusively with Thermoproteota order Sulfolobales have been reported to grow aerobically on $H_2$ as the sole electron donor but at supra-micromolar concentrations[31–34]. The enzymes responsible for this process were not resolved. A study on the thermoacidophile *Metallosphaera sedula* (Sulfolobales) demonstrated a [NiFe]-hydrogenase containing two hypothetical genes (*isp1, isp2*) was upregulated during aerobic autotrophic growth[35], though prior studies have suggested this enzyme is oxygen-sensitive, and instead, involved in anaerobic sulfidogenic growth[36,37]. In light of recent discoveries in bacteria, it also remains to be tested whether archaea are capable of consuming $H_2$ gas at sub-micromolar concentrations. Aerobic CO oxidation has been observed in thermoacidophilic (Sulfolobales)[38,39] and halophilic (Halobacteria)[40,41] archaea, the latter to sub-atmospheric levels[40,41]. While genomic and biochemical evidence suggests that this process is mediated by form I CO dehydrogenases, the physiological role of this process remains unresolved. Since atmospheric substrates are readily available to archaea living in oxic environments, a reasonable hypothesis is that some aerobic archaea use atmospheric $H_2$ and CO to conserve energy during growth and survival akin to bacteria.

Here we studied the aerobic thermoacidophilic archaeon *Acidianus brierleyi* (DSM 1651$^T$), which was isolated from an acidic hot spring in 1973 and nominally grows on heterotrophic substrates, mineral sulfides, reduced sulfur species and/or ferrous iron[42]. This organism also grows lithoautotrophically on micromolar levels of $H_2$ using either elemental sulfur under anoxic conditions or oxygen under oxic conditions as terminal electron acceptors[30,43]. Its capacity to use CO as a substrate has not been tested. Consistent with the hyper-thermoacidophilic growth of other members within Sulfolobales, *A. brierleyi* grows between 45 to 75 °C ($T_{opt}$ 70 °C) at pH values of 1 to 6 (pH$_{opt}$ 1.5 to 2.0)[43]. Previous studies have demonstrated that geothermal isolates from the phyla Acidobacteriota[6,44], Chloroflexota[45], Firmicutes[46], and Verrucomicrobiota[47] can each meet their energy needs using atmospheric $H_2$. We therefore hypothesized that some $H_2$-oxidizing archaea from such habitats, as represented by *A. brierleyi*, also use this gas. In this study, we characterized the kinetics, threshold, and temperature dependence of aerobic $H_2$ uptake in *A. brierleyi* using ultra-sensitive gas chromatography and investigated if nanomolar levels of $H_2$ bolster the survival of carbon-starved cells. We also tested whether another hydrogenotrophic and thermoacidophilic archaeal species, *Metallosphaera sedula* (DSM 5348$^T$), could consume atmospheric $H_2$. To identify the enzymes mediating this process, we analyzed the phylogeny, genetic organization, and predicted protein complex structures of the hydrogenases of *A. brierleyi*. Finally, we performed quantitative proteomics analysis of *A. brierleyi* cells grown at various growth stages with different electron acceptors to substantiate these inferences and gain insights into the ecophysiological roles of $H_2$ oxidation.

## Results

### *Acidianus brierleyi* rapidly oxidizes atmospheric $H_2$ and CO across a range of temperatures to support growth and survival

To determine if *A. brierleyi* is a high-affinity $H_2$ oxidizer, we grew triplicate cultures organoheterotrophically in closed serum vials to mid-exponential growth phase (OD$_{600}$ ~0.06; Fig. 1A), amended the vials with a dilute initial concentration of $H_2$ (~10 ppmv / ~6 nM), and monitored $H_2$ oxidation *via* gas chromatography. The cultures aerobically oxidized $H_2$ in an apparent first-order kinetic process, reaching a sub-atmospheric threshold concentration of 0.20 to 0.25 ppmv (0.12 to 0.16 nM) within 24 h (Fig. 1B). Concomitantly, there was no decrease of $H_2$ in heat-killed cells and sterile medium, validating the observation is due to archaeal activity (Fig. 1B, Supplementary Dataset 1). Likewise, the cultures consumed $H_2$ present at ambient levels (~0.62 ppmv in lab air), confirming atmospheric uptake was not stimulated by elevated $H_2$ (Fig. S1). Next, we measured $H_2$ uptake kinetics of *A. brierleyi* whole cells during aerobic conditions. $H_2$ uptake followed Michaelis-Menten kinetics ($R^2 = 0.90$) with an apparent half-saturation constant ($K_{m(app),70\,°C}$) of 3.67 μM (2.24–6.54, 95% confidence interval) and an apparent maximum oxidation rate ($V_{max(app),70\,°C}$) of 7.63 mmol g$_{protein}^{-1}$ h$^{-1}$ (6.2–10.1, 95% confidence interval), respectively (Fig. 1C). It should be noted that these parameters potentially reflect the combined activity of two or more kinetically distinct hydrogenases within *A. brierleyi* cells, given hydrogenases with micromolar affinity constants typically cannot oxidize atmospheric $H_2$[5]. $V_{max(app),70\,°C}$ and $K_{m(app),70\,°C}$ for $H_2$ uptake are higher during aerobic than sulfur-dependent anaerobic growth (Fig. 1D), suggesting this archaeon uses $H_2$ more quickly under aerobic conditions.

We then aerobically incubated mid-exponential and stationary phase cultures of *A. brierleyi* at 70 °C with ~10 ppmv each of $H_2$, CO, and methane (CH$_4$; as an internal standard) in the ambient air headspace. Throughout the time course, there was minimal gas leakage reflected by CH$_4$ concentrations within samples and controls (Fig. S2). A simultaneous decrease of headspace $H_2$ and CO was observed in cultures under both conditions (Fig. 2A), confirming that the CO dehydrogenase is active and suggesting aerobic oxidation of both gases is a constitutive process. Whereas $H_2$ was rapidly consumed to sub-atmospheric levels, CO was more slowly consumed to a supra-atmospheric steady-state concentration (1.3–2.1 ppmv). However, negative controls of sterile medium and killed cultures showed a linear and substantial production of CO (~2.0 ppmv day$^{-1}$, $R^2 = 0.99$) (Fig. 2A), likely due to thermal degradation of the butyl-rubber stopper or organic substrates in the medium as previously observed[48,49]. In order to mitigate this effect and establish an uptake threshold for CO, we incubated cultures at 37 °C where abiotic CO production was 20 times slower (~0.10 ppmv day$^{-1}$, $R^2 = 0.50$) (Fig. 2B). The cells co-consumed $H_2$ and CO during the time course to below atmospheric levels (Fig. 2B). Based on this result, Thermoproteota is the second archaeal phylum shown to be capable of atmospheric CO oxidation, following the Halobacteriota[40,41]. To determine if the species can oxidize trace gases at temperatures representative of temperate environments, $H_2$ oxidation experiments were carried out at 25 °C and 10 °C. As with the 37 °C experiments, cultures were first grown at 70 °C to the desired density, then equilibrated and incubated at colder temperatures to measure gas consumption. Atmospheric $H_2$ uptake was observed even two weeks after harvesting cultures (Fig. 2C), suggesting this process is highly resilient to temperature variations. Thus, an obligate thermophile *A. brierleyi* ($T_{opt}$ 70 °C), continually harvests trace gases at temperatures outside its growth ranges ($T_{min}$ 45 °C)[42,43].

We investigated if the metabolism of nanomolar $H_2$ enhances survival of *A. brierleyi* during carbon starvation. The viability of stationary phase cultures supplemented daily with a headspace of 0, 0.6 (ambient level in air), 5, and 50 ppmv $H_2$ was monitored during long-term incubations at both 70 °C and 25 °C. Optical density and cell protein declined more rapidly in persisting cultures at 70 °C than at

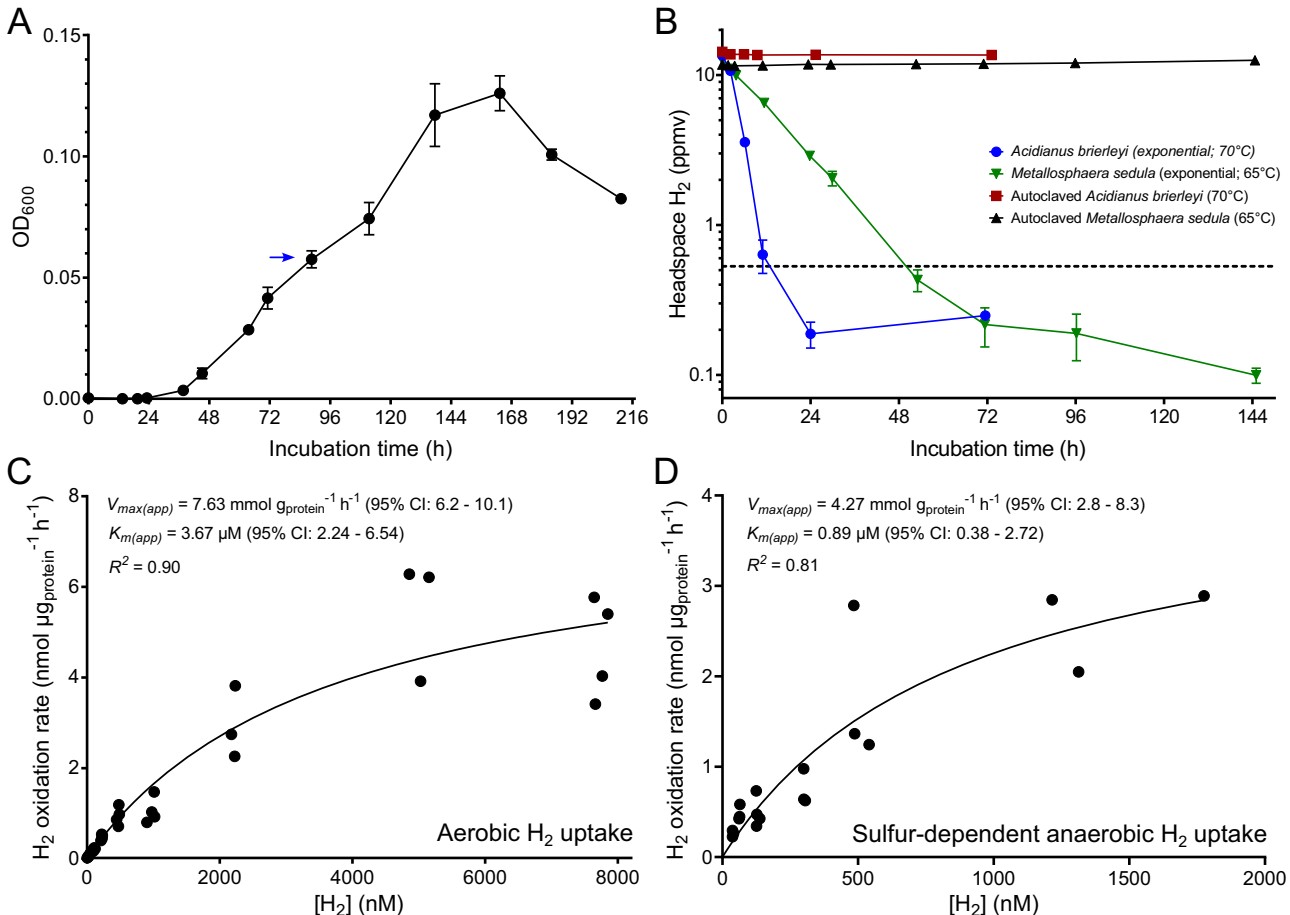

**Fig. 1 | Aerobic hydrogen (H₂) oxidation during exponential growth of *Acidianus brierleyi* at 70 °C. A** Heterotrophic growth of *A. brierleyi* in DSMZ medium 150 base with 0.2 g l⁻¹ yeast extract at 70 °C and pH 2. The blue arrow indicates the initial cell density where the H₂ consumption experiment in (**B**) was performed on *A. brierleyi*. (**B**) Gas chromatography measurement of H₂ oxidation to subatmospheric levels by *A. brierleyi* and *Metallosphaera sedula* at mid-exponential growth, with heat-killed cells as negative controls. Headspace H₂ mixing ratio is presented on a logarithmic scale and the dotted line indicates the mean atmospheric H₂ mixing ratio (0.53 ppmv). Data in both (**A**) and (**B**) are presented as mean ± S.D. values of three biological replicates (*n* = 3). Apparent kinetics of H₂ oxidation by mid-exponential *A. brierleyi* cultures grown heterotrophically in (**C**) oxic and (**D**) anoxic conditions with elemental sulfur as the electron acceptor. A Michaelis–Menten non-linear regression model was used to calculate kinetic parameters and derive a best fit curve.

25 °C, reflecting increased cell lysis and a substantial energetic cost to maintain cell integrity in hot acidic conditions (Fig. S3). ATP levels, culture density, and cell protein concentrations varied between the incubations in a manner that reflects H₂ availability (Fig. S3). Effects were modest during incubations at 70 °C: after 20 days of persistence, cellular ATP (nmol mg$_{protein}$⁻¹) of cultures supplemented with 50 ppmv H₂ was significantly greater than without H₂ by 27% at 70 °C ($p < 0.05$) (Fig. 2D); there was no significant difference for 0.6 and 5 ppmv treatment compared to zero H₂ control, though ATP levels, optical density, and cell protein are generally higher (Fig. S3). H₂ supplementation at 25 °C greatly enhanced cell viability. Cellular ATP was significantly greater across all H₂ treatments, including at atmospheric level, after 10 days of persistence (Fig. 2E); at day 20, cultures supplemented with 0.6, 5, and 50 ppmv H₂ had 38%, 129%, and 672% greater cellular ATP than the control, and were greater compared to time zero for cultures provided with 50 ppmv H₂ (Supplementary Dataset 2). Concordant but weaker patterns were observed for culture density and cell protein measurements (Fig. S3). These results strongly suggest *A. brierley* conserves energy from aerobic oxidation of trace H₂ at nanomolar ranges during persistence and atmospheric substrates are significant for cells to stay energized at temperate conditions.

To extend these findings to other aerobic archaea, we tested trace gas oxidation by another hydrogenotrophic Sulfolobales species, *M. sedula*. This archaeon consumed headspace H₂ at a slower rate than *A.*

*brierleyi*, but reached a lower sub-atmospheric consumption threshold (~0.1 ppmv / 0.06 nM) during exponential growth (OD$_{600}$ ~ 0.047) at 65 °C (Fig. 1B). This archaeon did not consume CO, an observation consistent with its genome lacking a CO dehydrogenase. Together, these results provide the first evidence that atmospheric H₂ oxidation is not exclusive to bacteria, but rather also extends to the archaeal domain.

### *Acidianus brierleyi* possesses four phylogenetically and syntenically distinct [NiFe]-hydrogenases widely distributed in Sulfolobales

We built an HMM profile using all hydrogenase large subunit reference sequences from the hydrogenase database (HydDB)[50] to comprehensively search for the enzymes responsible for H₂ oxidation in *A. brierleyi*. Through careful inspection of hits, we identified four [NiFe]-hydrogenases encoded by this organism. They belong to uptake group 1 and 2 [NiFe]-hydrogenases but share < 30% sequence similarity with each other. To contextualize their relationships, a phylogenetic tree was constructed of the catalytic subunits of these hydrogenases, HydDB references, and newly identified group 1 and 2 [NiFe]-hydrogenases from all representative archaeal species in the Genome Taxonomy Database (GTDB) release 202[51] (Fig. 3 and Supplementary Dataset 5). The four hydrogenases clustered into distinct lineages. DFR85_RS19635 (HcaL) is a member of the previously defined 1g

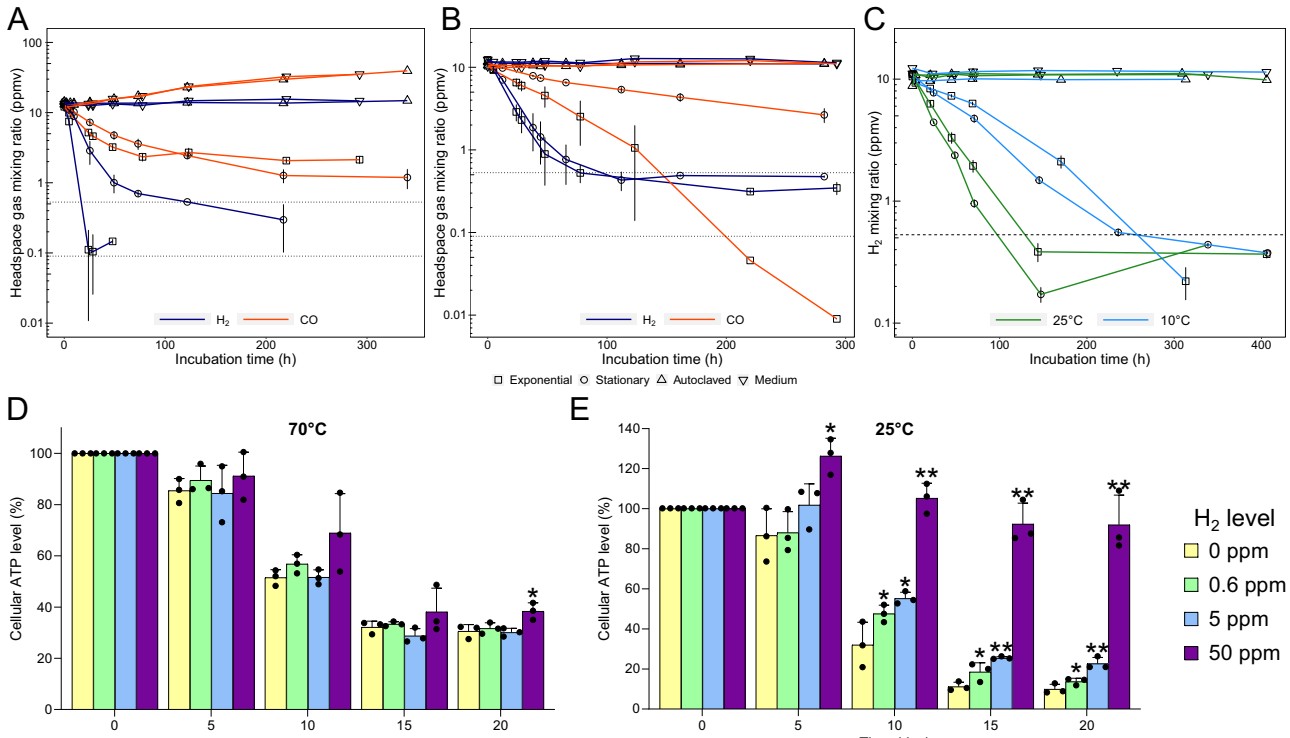

**Fig. 2 | *Acidianus brierleyi* consumes atmospheric trace gases to enhance survival at various temperatures.** Gas chromatography measurement of simultaneous consumption of $H_2$ and CO at (**A**) 70 °C and (**B**) 37 °C; and (**C**) continued $H_2$ oxidation activity at 25 °C and 10 °C by mid-exponential and stationary phase cultures of *A. brierleyi*. Relative change in cellular ATP level (nmol/mg$_{protein}$) of stationary phase cultures of *A. brierleyi* supplemented daily with low levels of headspace $H_2$ within twenty days at (**D**) 70 °C and (**E**) 25 °C. For all panels, data are presented as mean ± S.D. values of three biological replicates ($n = 3$); dotted lines in (**A**–**C**) indicate the mean global atmospheric $H_2$ (0.53 ppmv) and CO (0.09 ppmv) mixing ratios; and asterisks in (**D**–**E**) indicate significantly higher cellular ATP level in $H_2$-supplemented cultures than cultures with no extra $H_2$ at the same timepoint based on one-sided Student's $t$ test (*$p < 0.05$; **$p < 0.005$). The exact $p$ values are reported in the Source Data file.

subgroup[18], also known as Crenarchaeaota Isp-type hydrogenase[52], characterized by the presence of genes encoding a putative transmembrane *b*-type cytochrome (*isp1*) and iron-sulfur protein (*isp2*) between the small and large subunit genes. This unique genetic arrangement is conserved in *A. brierleyi* (Fig. 4A). This enzyme is the sole biochemically and genetically characterized uptake [NiFe]-hydrogenase from Sulfolobales and is known to interact with sulfur reductase to mediate sulfidogenic $H_2$ oxidation[36,37,53–55]. They are also the only formally reported hydrogenase from *Acidianus*, and thus often assumed to mediate aerobic $H_2$ oxidation[35]. DFR85_RS29945 (HysL) clusters with subgroup 2e, a putative hydrogenase lineage defined through genome surveys[50] and only previously reported in *M. sedula*[35]. The gene cluster predicted to encode the structural subunits of this enzyme (*hysLSM*) is similar to the high-affinity group 2a enzyme from bacteria (Fig. 4A)[56].

The two other deep-branching hydrogenases are from previously unidentified subgroups exclusive to Sulfolobales. Designated novel Sulfolobales clade SUL1 (HsuL1) and SUL2 (HsuL2) hereafter, they have the highest sequence identity with the group 1l (31.4% identity; WP_080561888.1 - *Sulfolobus islandicus*) and 1h hydrogenases (29.3%; WP_015922819.1 - *Thermomicrobium roseum*) in HydDB respectively. SUL1 is encoded by a lone pair of large and small subunits. In contrast, the small and large subunit genes of SUL2 are immediately followed by *isp1* and *isp2* genes (Fig. 4A). This observation is unexpected given the *isp* genes were thought to be exclusive to two distantly related hydrogenase subgroups (bacterial 1e and archaeal 1g) and always located between two hydrogenase structural subunits[18,37,52,57]. None of the four [NiFe]-hydrogenases affiliate directly with subgroups known to mediate atmospheric $H_2$ oxidation (i.e. groups 1h, 1l, 1f, 2a), suggesting unique lineages catalyze this activity in *A. brierleyi*. Other novel

uncharacterized hydrogenase subgroups were also widespread in other uncultured aerobic and anaerobic archaea. At least three other divergent and previously undescribed group 1 [NiFe]-hydrogenase clades, spanning Altarchaeota, Halobacteriota, Hydrothermarchaeota, Thermoproteota, and Thermoplasmatota, were identified (Fig. 3 & Supplementary Dataset 5).

To gain insight into the structure, subunit composition, and physiological functions of the four *A. brierleyi* hydrogenases, we modeled their structures using AlphaFold2 Multimer[58,59]. The results of this modeling are discussed in detail in Supplemental Note 1 and summarized here. The models of both Hca (group 1g) and Hsu2 (clade SUL2) consist of extracellular large and small [NiFe]-hydrogenase subunits, which form a membrane-spanning complex with the integral membrane *b*-type cytochrome protein Isp1 associated with the multi iron-sulfur protein Isp2 on the cytoplasmic side of the membrane (Fig. 4B, C). This subunit arrangement gives rise to an electron transport relay that may allow both hydrogenases to transfer electrons from $H_2$ oxidation to membrane-bound quinone via a heme group of Isp1 and to a lower potential electron acceptor (e.g. NAD$^+$ or ferredoxin) in the cytoplasm via Isp2 (Fig. 4B, C). This arrangement would thus enable these hydrogenases to provide electrons for both maintaining the proton-motive force via quinone oxidation (by either sulfur reductase or terminal oxidases), as well as driving biosynthesis and carbon fixation during autotrophic growth. Modeling of Hys (group 2e) reveals an unexpected similarity to the recently characterized group 2a hydrogenase Huc from *Mycobacterium smegmatis*, which oxidizes $H_2$ at sub-atmospheric concentrations and forms a large oligomer around the central membrane-associated subunit HucM[60]. Despite limited sequence identity ( ~ 45%), modeling reveals the HysSL subunit forms a dimer, which is highly similar to that of HucSL

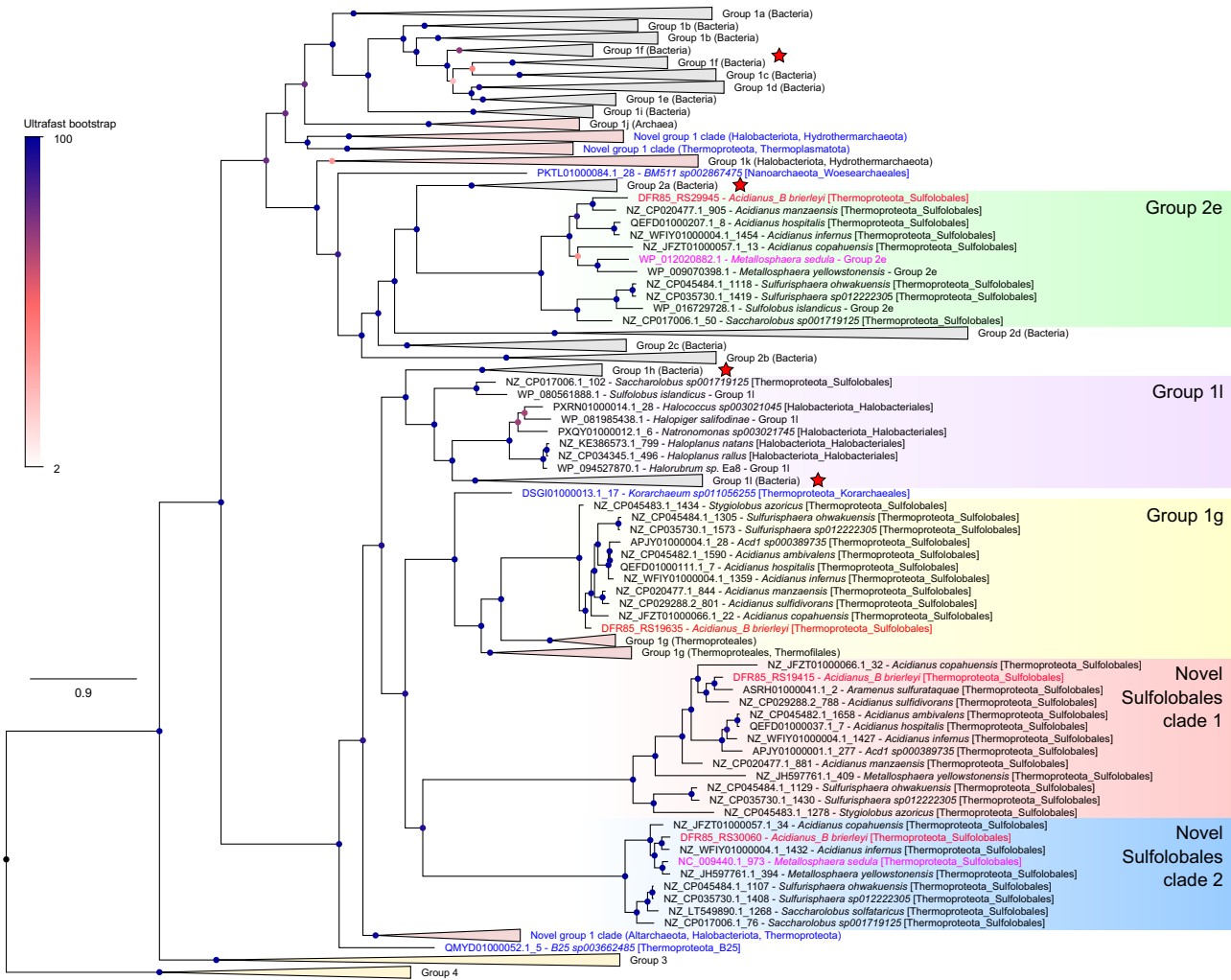

**Fig. 3 | Identification of four [NiFe]-hydrogenases in *Acidianus brierleyi*.**
Maximum-likelihood phylogenetic reconstructions of amino acid sequences of the uptake group 1 and 2 [NiFe]-hydrogenase large (catalytic) subunits identified in *A. brierleyi* (4 sequences; red text), *M. sedula* (2 sequences; fuschia text), genomes of all archaeal representative species in Genome Taxonomy Database (GTDB) release 202 (202 sequences)[51], and hydrogenase reference database HydDB (1003 sequences)[50]. Group 3 and 4 [NiFe]-hydrogenases were included as out-groups and the phylogeny was rooted between group 4 [NiFe]-hydrogenases and all other groups. Note that *A. brierleyi* is classified under a distinct genus from *Acidianus* (placeholder name *Acidianus_B*) in GTDB. Collapsed subgroups/clades that were exclusively bacterial or archaeal are shaded in gray or pink, respectively. Novel archaeal hydrogenase lineages are colored in blue or specified otherwise. Star symbols denote hydrogenase subgroups with members experimentally shown to mediate atmospheric $H_2$ oxidation. Details on alignment and tree inference can be found in Methods and all sequences are provided in Supplementary Dataset 3. Each node was colored by ultrafast bootstrap support percentage (1000 replicates) and the scale bar indicates the average number of substitutions per site. The tree showing all taxa is provided in Supplementary Dataset 5.

(RMSD = 1.55 Å; Fig. 4D). In a similar manner, the modeled structure of HysM is a homolog to HucM (despite only sharing 18% sequence identity) and forms a characteristic tube-like structure, which in Huc scaffolds the enzyme and delivers quinone to the electron acceptor site of HucS[60]. Based on these similarities, we generated a full model for Hys (Fig. 4E). Finally, the model for Hsu1 (clade SUL1) reveals a truncated HsuS1 subunit with a single [FeS]-cluster (Fig. S4A). Hsu1 may form a complex with an integral membrane quinone-reducing complex related to succinate dehydrogenase, encoded by genes directly upstream of HsuS1 (Fig. S4B).

Through analysis of the *A. brierleyi* genome, we also identified an operon encoding a form I CO dehydrogenase responsible for the observed carbon monoxide oxidation (Fig. S6). It has a typical genetic arrangement of form I CO dehydrogenase (*coxEDLSMF*), and its large subunit (*coxL*) harbors the conserved AYXCSFR active site signature motif[21]. Phylogenetic analysis of the CoxL protein shows that the *A. brierleyi* enzyme clusters with those from diverse bacteria and other thermoacidophilic archaea, whereas halophilic archaea possess a distinct deeper-branching enzyme (Fig. S7)[22]. Thus, members of

Sulfolobales likely acquired CO dehydrogenase from aerobic bacteria through horizontal gene transfer and independently of halophilic archaea.

## SUL2 is a membrane-associated complex that is the most abundant and active hydrogenase under aerobic conditions

To investigate and characterize hydrogenases responsible for the aerobic $H_2$ uptake, proteins from *A. brierleyi* cytosolic and membrane fractions were separated using blue native polyacrylamide gel electrophoresis, followed by hydrogenase activity staining with the artificial electron acceptor nitrotetrazolium blue (NBT) incubated in an anaerobic atmosphere with 7% v/v $H_2$. In membrane fractions of cells grown on ambient levels of $H_2$, hydrogenase activity was detected in a faintly stained band with an apparent molecular size between 146–242 kDa (Fig. 5A). Bands corresponding to the same molecular size stained at higher intensity in membrane and cytosolic fractions of cells grown on 20% v/v $H_2$ in air; an additional band in the membrane fraction with apparent size ~480 kDa was also observed in this growth condition (Fig. 5A).

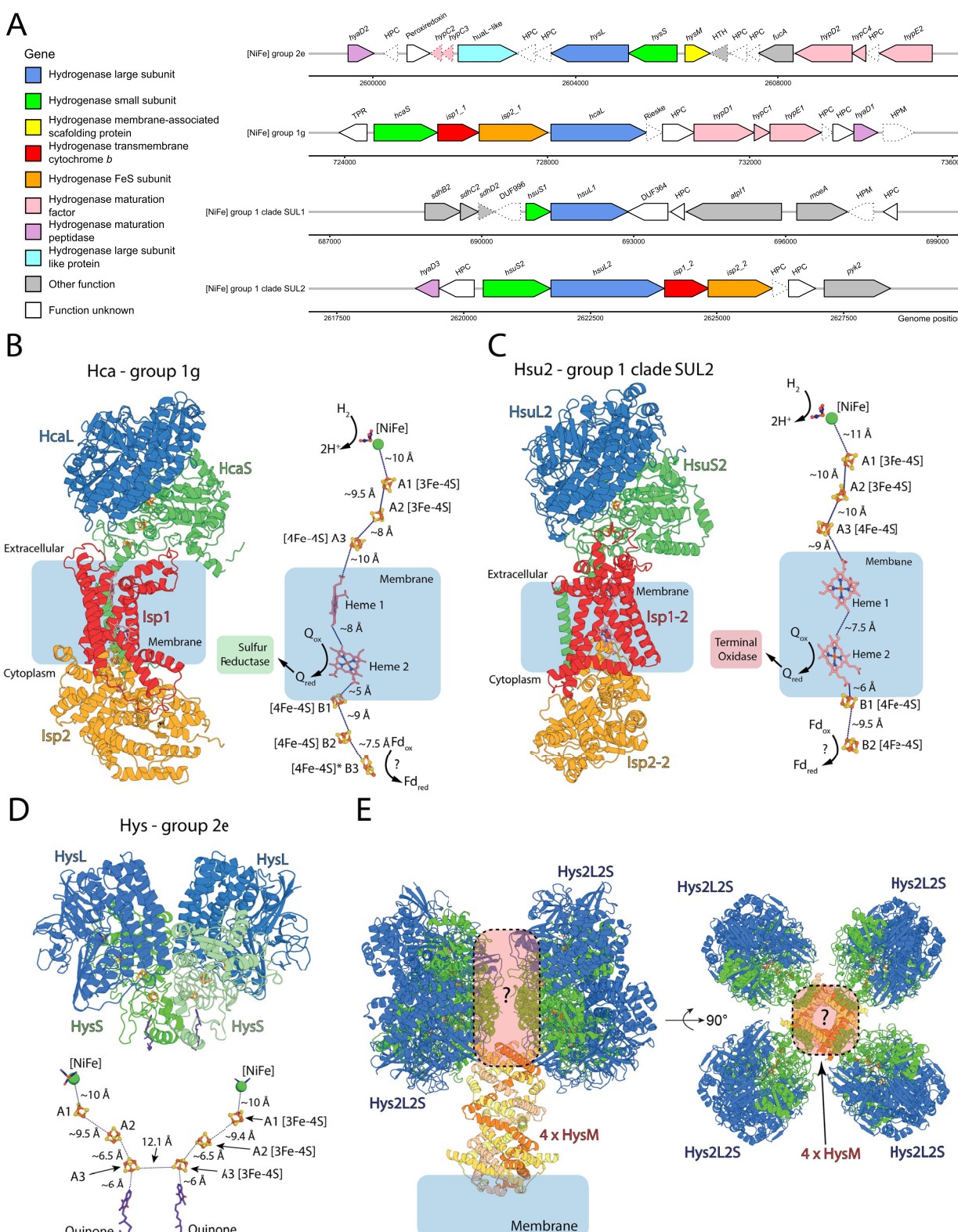

**Fig. 4 | Genetic organization and AlphaFold structural models of the four [NiFe]-hydrogenases in *Acidianus brierleyi*. A** Genetic organization of the four [NiFe]-hydrogenases encoded by *A. brierleyi*. Arrow outlines denote the presence (solid line) and absence (dotted line) of protein expression of the gene detected by shotgun proteomics under tested conditions. Gene length is shown to scale. HPC, hypothetical cytosolic protein; HPM, hypothetical membrane protein. Detailed information on loci, annotations and amino acid sequences of each gene are available in Supplementary Dataset 4. AlphaFold-derived models of the group 1g

hydrogenase Hca (**B**), the novel group 1 clade SUL2 hydrogenase Hsu2 (**C**), and the group 2e hydrogenase Hys (**D**) are shown as a cartoon representation of the complex formed by the subunits identified in panel A (left/top panel) and the electron transport relay formed by modeled cofactors, with predicted electron donors and acceptors indicated (right/bottom panel). **E** A model of a higher order complex formed by Hys, incorporating the AlphaFold model of the HysM subunit, and based on the Cryo-EM structure of the group 2a hydrogenase Huc from *Mycobacterium smegmatis*[60].

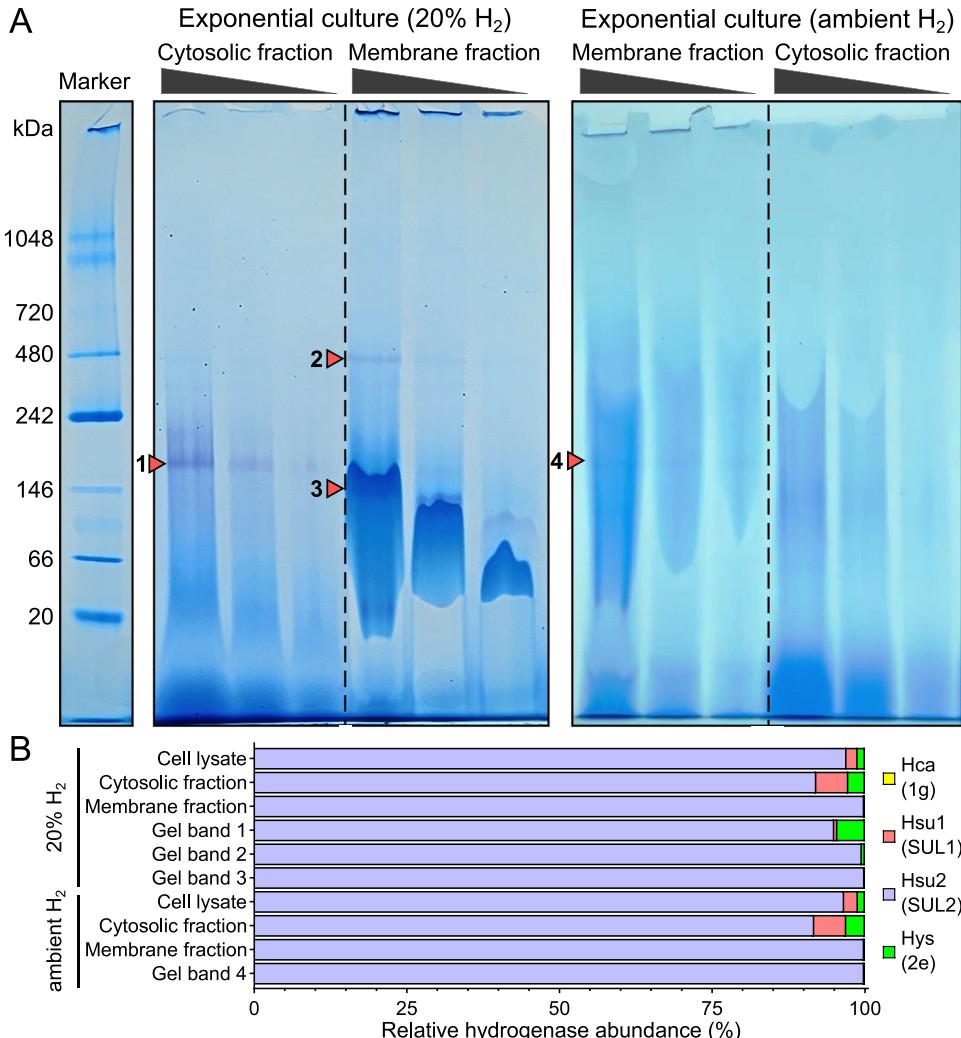

**Fig. 5 | Zymographic and mass spectrometric detection of hydrogenase activity in aerobic culture of *Acidianus brierleyi*. A** Hydrogenase activity staining of cytosolic and membrane fractions of *A. brierleyi* cells separated by blue native polyacrylamide gel electrophoresis and stained with artificial electron acceptor nitrotetrazolium blue chloride (NBT) under a 7%-$H_2$ anaerobic atmosphere. Cells were harvested during aerobic organotrophic exponential growth in a headspace with either 20% $H_2$ or ambient $H_2$. 10, 5, and 2.5 μl of each fraction were loaded on the gel. The protein ladder in the left lane was stained with Coomassie blue. Purple bands indicated by arrows suggest reduction of NBT by hydrogenase activity. The shift in bands in membrane fraction is a known phenomenon due to the interaction between lipid, small membrane proteins, and Coomassie Blue G-250[108]. The same experiment was repeated independently twice with same results (*n* = 2). (**B**) Mass spectrometric identification hydrogenases in *A. brierleyi* cell lysate, cytosolic fraction, membrane fraction, and positive bands indicated in (**A**). Relative abundance of the four hydrogenases based on iBAQ (Intensity-Based Absolute Quantification) is shown.

Protein mass spectrometry was used to identify and quantify relative abundance of the four putative hydrogenases in the various cell fractions and gel bands showing hydrogenase activity (Supplementary Dataset 4). The SUL2 enzyme accounted for ~97%, ~92%, and ~99.9% of total hydrogenases detected in cell lysate, cytosolic fraction, and membrane fraction, respectively. The group 2e and SUL1 hydrogenases were enriched in the cytosolic fraction, but comprised just ~3% and ~5% of total hydrogenases (Fig. 5B). These observations corroborate with our AlphaFold2 predictions that group 2e and SUL1 hydrogenases interact transiently with the membrane whereas Hsu2 forms a tighter membrane complex (Fig. 4C–E; Fig. S4). The group 1g hydrogenase was not detected in any of the samples. In gel bands displaying hydrogenase activity, the four subunits of SUL2 hydrogenase were identified and predominated (Fig. 5B). Consistently, the predicted molecular weight of SUL2 complex (211 kDa) is in line with the apparent size range of gel bands 1–3 shown in Fig. 5b, though activity in gel band 4 may be explained by oligomers or super-complexes of SUL2 hydrogenase. No specific gel bands corresponding to the group 2e and SUL1 hydrogenases were detected, likely reflecting their low abundance.

Together, these results reveal that the SUL2 hydrogenase is active and preferentially membrane-localized, and it is probably the primary enzyme for aerobic $H_2$ oxidation in *A. brierleyi*.

### Quantitative proteome comparison identifies differentially regulated hydrogenases in response to principal growth substrates and terminal electron acceptors

To gain a system-wise understanding of how various hydrogenases contribute to *A. brierleyi* physiology and energetics, quadruplicate shotgun proteomes were compared for cells grown at four different conditions: mid-exponential growth on heterotrophic medium (EX); stationary phase on heterotrophic medium (ST); sulfur-dependent anaerobic hydrogenotrophic growth ($H_2$:$CO_2$:$N_2$ = 20%:5%:75% v/v) on mineral medium (AN); and aerobic hydrogenotrophic growth ($H_2$:$CO_2$:air = 20%:5%:75% v/v) on mineral medium (AE). We identified a total of 1847 proteins (~65% of protein-coding genes) falling below a predefined false discovery rate threshold of 1%. Principal component analysis showed that replicates were highly similar, but each sample group was distinct from one another (Fig. S8).

Proteome composition reflected the availability of the principal energy source and cellular status under each condition (Fig. 6, Supplementary Dataset 4). Proteins involved in amino acid degradation, the Entner-Doudoroff glycolytic pathway, and the tricarboxylic acid cycle for heterotrophic metabolism were highly abundant in heterotrophically grown cells (EX, ST) compared to cells at autotrophic conditions (AN, AE). For example, indolepyruvate:ferredoxin oxidoreductase subunit alpha (IorA) for amino acid degradation, 2-dehydro-3-deoxygluconokinase (KDGK) in the non-phosphorylative Entner-Doudoroff pathway, citrate synthase (GltA2), and succinate dehydrogenase flavoprotein subunit (SdhA) in the tricarboxylic acid cycle were significantly more abundant (6.7–21.6, 1.4–10.7, 3.2–5.5, and 1.8–2.2 fold respectively) under heterotrophic conditions (adj. $p < 0.01$) (Fig. 6). Contrastingly, cells under autotrophic conditions increased synthesis of proteins involved in 4-hydroxybutyrate cycle for carbon fixation and acetyl-CoA assimilation (adj. $p < 0.001$), including acetyl-CoA/propionyl-CoA carboxylase (AccADBC; 6–30 fold) and 4-hydroxybutyrate-CoA ligase (HbsC; 3.6–10.7 fold) (Fig. 6, Supplementary Dataset 4). Furthermore, marker proteins involved in cell division (e.g. CdvA, Cdc48), translation (e.g. RLI, IF1A, EF2), amino acid biosynthesis (e.g. HisD, MetE, IlvC1), and nucleic acid biosynthesis (e.g. GuaA2, PyrG) were significantly decreased in stationary phase cultures in comparison to the other three actively growing cultures (adj. $p < 0.01$) (Fig. 6). Consistent with the activity-based measurements (Fig. 2), CO dehydrogenase subunits (CoxS, CoxM, CoxL) were produced at moderate levels during aerobic heterotrophic growth and high levels (average four-fold increase) during stationary phase (Supplementary Dataset 3). This is reminiscent of observations that CO dehydrogenase is upregulated and active during starvation in various bacterial cultures[22,45,61–63]. Stationary phase cells also synthesized a greater abundance of extracellular solute-binding proteins (e.g. DppA1, DppA2, ESB2) and major facilitator superfamily sugar/acid transporters (MSAT) (Fig. 7, Fig. S10), likely enhancing scavenging of trace carbon substrates during energy limitation. Altogether, these results suggest that proteomics provides a reliable representation of cellular metabolism.

For proteins involved in H2 metabolism, the regulation of the group 1g [NiFe]-hydrogenase is in line with its reported role in sulfur-dependent hydrogenotrophy. Subunits of this hydrogenase (HcaS, Isp2_1, Isp1_1) and sulfur reductase (SreA, SreB, SreC) were among the ten most upregulated proteins during anaerobic autotrophic growth (adj. $p < 0.001$) (Fig. 6, Supplementary Dataset 4). These proteins were otherwise undetected in the three oxic conditions (aside from negligible abundance in a single AE replicate), in line with the activity staining results (Fig. 5). HcaSL represented 87.5% of all hydrogenase large and small subunits during anaerobic autotrophic growth, but had the lowest abundance during aerobic autotrophic growth. Hydrogenase maturation factors (*hypD1*, *hypC1*, *hypE1*) and proteases (*hyaD1*) neighboring this hydrogenase shared a similar expression profile (Fig. S9). Expression of the hydrogenase lineage SUL1 was similar to Hca, as HsuL1 and HsuS1 had their greatest abundance under anaerobic autotrophic growth, though they only constituted 3.3% of hydrogenase proteins at this condition (Fig. 6). In contrast, the hydrogenase lineage SUL2 displayed a reverse expression pattern, in which its abundance peaked during aerobic hydrogenotrophic growth (Fig. 6, Fig. 7), compared to the three other conditions (adj. $p < 0.001$) (Fig. 6, Supplementary Dataset 4). In agreement with the activity-based studies (Fig. 5), SUL2 was the only hydrogenase that was found to be constantly and copiously produced during all four tested conditions. HsuL2 and HsuS2 represented 58.4%, 79.2%, 9.1%, and 97.6% of all quantified hydrogenase large and small subunits during exponential heterotrophic, stationary heterotrophic, anaerobic autotrophic, and aerobic autotrophic conditions respectively (Supplementary Dataset 4). Comparing expression in heterotrophically grown cultures, this hydrogenase was upregulated in the carbon-limited stationary phase

condition in a manner reminiscent of most high-affinity group 1h [NiFe]-hydrogenases studied in bacteria[24]. Finally, the group 2e hydrogenase was least abundant among the four hydrogenases in all tested conditions (Fig. 6). Contrary to prior speculations, it is therefore unlikely to be involved in aerobic hydrogenotrophic growth[50], though may still be expressed at sufficient levels to scavenge atmospheric H2.

## Discussion

In this work, we report that two aerobic archaea scavenge H2 during growth and starvation, including at sub-atmospheric levels. These observations expand the guild of atmospheric H2 oxidizers to include domain archaea and suggest that members of this domain contribute to the climate-relevant atmospheric H2 sink. The aerobic oxidation of H2 by *A. brierleyi* follows Michaelis-Menten kinetics with a high $K_{m(app),70\,°C}$ in micromolar ranges of H2, but a low uptake threshold in the picomolar range, highlighting that the organism efficiently uses H2 at a very broad range of concentrations. The constitutive production and kinetic properties of the hydrogenases of *A. brierleyi* likely confer competitive advantages in dynamic geothermal habitats. Cells can rapidly mobilize H2 available at elevated concentrations, for example, from volcanic efflux and venting steam, or at oxic-anoxic interfaces where fermentative H2 is produced and diffuses from deeper anoxic layers[64–66], to fuel autotrophic or mixotrophic growth. During periods when substrates are variable and limiting, the ability to harvest dilute concentrations of H2 either from the source environment or the atmosphere, for example, in the immediate vicinity of air-sediment or air-water interfaces, will enable them to meet maintenance energy requirements. This supplemental energy source can be particularly beneficial for organisms living in acidic and heated environments, as continuous energy expenditure is required to maintain a neutral intracellular pH against a steep external proton gradient and synthesize defence biomolecules to protect against oxidative and thermal damage[67,68]. Indeed, carbon-starved cells supplied with dilute H2 (50 ppmv) were more viable after 20 days of persistence at 70 °C (Fig. 2D). Notably, the absence of a distinguishable viability increase at lesser concentrations was likely due to the substantial energy demand to maintain cell integrity under poly-extreme conditions. Apart from demonstrating *A. brierleyi* is an atmospheric H2 oxidizer, we additionally confirm that this archaeon is an atmospheric CO oxidizer, extending this metabolism to a second archaeal phylum (Thermoproteota). Reflecting CO dehydrogenase is most highly expressed in stationary phase cultures, this metabolism may primarily support survival during organic carbon starvation, in line with observations in bacteria[22,45,61–63].

While aerobic oxidation of H2 in Sulfolobales has been widely reported, it is likely that previous studies misattributed the enzyme responsible for this metabolism. Group 1g [NiFe]-hydrogenases are the only hydrogenases previously characterized in this order, with group 2e enzymes also reported based on comparative genomic analysis[18,35]. However, these enzymes are known to couple H2 oxidation with sulfur reduction (but not with oxygen), and are commonly present in obligately anaerobic archaea. In this study, we revealed the presence of two additional putative uptake [NiFe]-hydrogenases in *A. brierleyi* and the wider Sulfolobales order (Fig. 3), designated SUL1 and SUL2. Supported by the comprehensive proteomic and zymographic analysis, we introduce a model of H2 metabolism in heterotrophic and hydrogenotrophic growth of *A. brierleyi* (Fig. 7). Reflecting its high abundance during sulfur-dependent hydrogenotrophic growth, but minimal expression under oxic conditions, the group 1g [NiFe]-hydrogenase functions as the primary hydrogenase for H2 oxidation under anaerobic conditions and relays electrons to sulfur as the terminal electron acceptor. Given its overwhelming expression and activity during aerobic hydrogenotrophic growth, the hydrogenase SUL2 is the predominant hydrogenase for H2 oxidation under aerobic conditions and transfers electrons to oxygen as the terminal electron acceptor. In agreement with this notion, homologs of SUL2 and group

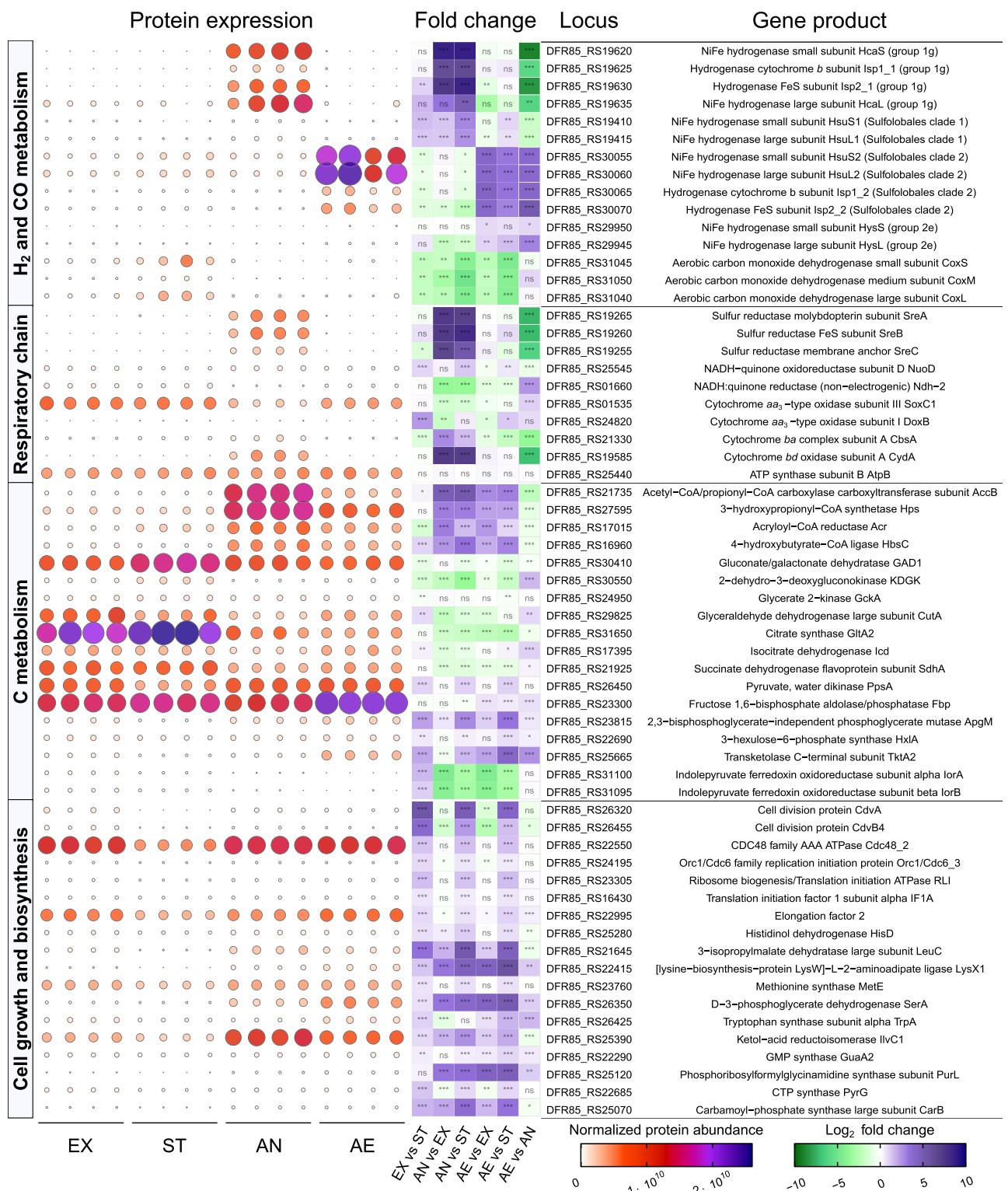

**Fig. 6 | Quantitative comparison of selected *Acidianus brierleyi* proteins under heterotrophic growth, stationary phase, sulfur-dependent hydrogenotrophic growth, and aerobic hydrogenotrophic growth.** Culture condition (four biological replicates each): EX, mid-exponential growth phase on heterotrophic medium; ST, stationary phase on heterotrophic medium; AN, anaerobic sulfur-dependent hydrogenotrophic growth; AE, aerobic hydrogenotrophic growth (Methods). Normalized protein abundance value represents MaxLFQ total intensity for the protein. Bubble size and color indicate protein abundance of the corresponding gene product in each biological replicate. Significant difference in fold changes of protein abundance of each condition pair is denoted by asterisks (two-sided Student's *t*-test with Benjamini-Hochberg correction; adjusted *p* value ≤ 0.001, \*\*\*≤0.01, \*\*≤0.05, \*>0.05, ns). The full set of quantitative proteomics results is provided in Supplementary Dataset 4.

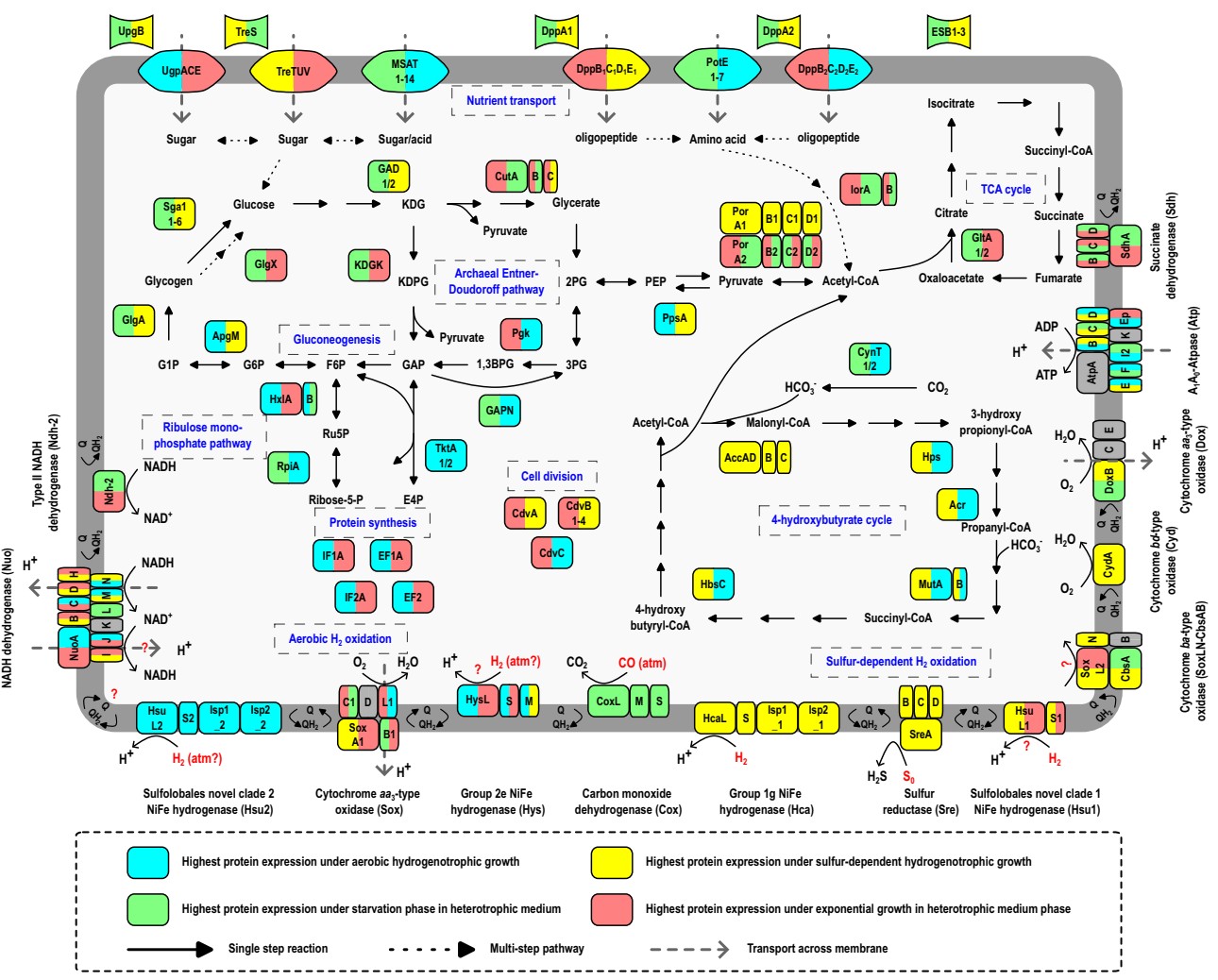

**Fig. 7 | Genome and proteome-based model of $H_2$ metabolism in heterotrophic and hydrogenotrophic growth of _Acidianus brierleyi_.** The color scale indicates the conditions where proteins had the highest expression (left) and second highest expression if their log2 fold difference is less two (right). Metabolic marker genes for central carbon metabolism, trace gas oxidation, and respiratory chain are shown. AccADBC, acetyl-CoA/propionyl-CoA carboxylase; Acr, acryloyl-CoA reductase; ApgM, phosphoglycerate mutase; CdvABC, cell division proteins; CutABC, glyceraldehyde dehydrogenase; CynT, carbonic anhydrase; DppBCDE, ABC di/oligopeptide transporter; EF1A/EF2; elongation factors; GAD, gluconate/ galactonate dehydratase; GAPN, glyceraldehyde-3-phosphate dehydrogenase; GlgA, glycogen synthase; GlgX, glycogen debranching protein; GltA, citrate synthase; HbsC, 4-hydroxybutyrate-CoA ligase; Hps, 3-hydroxypropionyl-CoA synthetase; HxlA, 3-hexulose-6-phosphate synthase; HxlB, 6-phospho-3-hexuloisomerase; IF1A/IF2A, translation initiation factors; IorAB, Indolepyruvate:ferredoxin oxidoreductase; KDGK, 2-dehydro-3-deoxygluconokinase; MSAT, major facilitator superfamily sugar/acid transporter; MutAB, methylmalonyl-CoA mutase; Pgk, phosphoglycerate kinase; PorABCD, pyruvate synthase; PotE, APC amino acid permease; PpsA, pyruvate, water dikinase; RpiA, ribose 5-phosphate isomerase; Sga1, glucoamylase; TktA, transketolase; UgpACE/TreTUV, ABC sugar transporter; UpgB/ TreS/DppA/ESB, extracellular solute-binding proteins. Note that _A. brierleyi_ is also known to grow chemolithoautotrophically on various reduced sulfur compounds and $Fe^{2+}$ (Fig. S10), as extensively described in previous studies[55,109,110].

2e hydrogenases are the sole hydrogenases identified in the obligately aerobic hydrogenotroph _M. sedula_[30] (Supplementary Dataset 5). Some electrons derived from both hydrogenases are potentially transferred to NADH dehydrogenase through reverse electron transport to generate the reductant necessary for carbon fixation via the 4-hydroxybutyrate cycle. Alternatively, as inferred from the AlphaFold2 structural model (Fig. 4), the SUL2 and group 1g enzymes may relay electrons from $H_2$ both to high-potential quinones through the Isp1 subunit for proton-motive force generation and low-potential acceptors (e.g. ferredoxin) through the Isp2 subunit for biosynthesis. This may occur through alternation of electron transport paths, for example in response to changes in cofactor ratios or through modularity in their structures, as previously described for the heliobacterial photosynthetic reaction center[69,70]. It also cannot be ruled out that these hydrogenases may simultaneously transfer electrons to both acceptors through the process of electron-bifurcation. Such an efficient energy conservation mechanism may explain the dominance of these hydrogenases during hydrogenotrophic growth. However, direct experimental validation using purified enzymes would be necessary to discriminate these possibilities.

Although zymographic and proteomic evidence strongly suggests hydrogenase SUL2 is responsible for aerobic oxidation of $H_2$ at high concentrations, it is unresolved whether this enzyme mediates atmospheric uptake. Based on our result that the obligately aerobic _M. sedula_ is a high-affinity $H_2$ oxidizer (Fig. 1B), it is logical to constrain the enzyme mediating atmospheric $H_2$ oxidation to the only two hydrogenases it encodes, namely the SUL2 or group 2e [NiFe]-hydrogenases (Fig. 3). SUL2 potentially mediates atmospheric $H_2$ oxidation based on four lines of evidence: (i) it is the only hydrogenase constantly detected and abundantly expressed in all tested conditions, in agreement with the constitutive activity of atmospheric $H_2$ oxidation in oxic conditions (Fig. 2); (ii) it shares the highest similarity with group 1h [NiFe]-hydrogenases known to mediate atmospheric $H_2$ uptake in bacteria, albeit at < 30% identity; (iii) the lineage is only present in aerobic or facultatively

aerobic Sulfolobales species (Fig. 3); and (iv) its high expression during stationary phase (compared to exponential phase) is analogous to the regulation of many group 1h [NiFe]-hydrogenases (Fig. 6)[24]. The micromolar whole-cell $H_2$ affinity constant is more consistent however with low-affinity [NiFe]-hydrogenases being the dominant $H_2$-metabolizing enzymes[5]. The group 2e [NiFe]-hydrogenase may alternatively account for atmospheric $H_2$ uptake. This reflects that this enzyme is a direct sister lineage and shares a highly similar predicted protein structure to the high-affinity bacterial group 2a hydrogenase (Fig. 3, Fig. 4), is solely present in aerobic or facultatively aerobic Sulfolobales species (Fig. 3), and is downregulated under anaerobic conditions (Fig. 6). Though it is expressed at relatively low levels under tested conditions, such levels may be sufficient for a catalytically efficient high-affinity hydrogenase to conserve sufficient energy from atmospheric $H_2$. Indeed, the wide kinetic range of *A. brierleyi* could be explained through the complementary activity of both low-affinity (e.g. SUL2) and high-affinity (e.g. group 2e) enzymes. The hydrogenase SUL1 is unlikely to account for atmospheric $H_2$ oxidation given its presence in the obligately anaerobic *Stygiolobus azoricus*[71] (Fig. 3), decreased abundance in oxic conditions (Fig. 6), and absence in *M. sedula*. Heterologous expression in genetically tractable thermoacidophiles such as *Sulfolobus acidocaldarius* and *Sulfolobus islandicus* offers a possible means to resolve the function of each hydrogenase[72].

Our results also show that CO and $H_2$ oxidation can co-occur at temperatures below growth ranges of extremely thermoacidophilic *A. brierleyi*. $H_2$ oxidation remained active at temperatures as cold as 10 °C for over two weeks, although future studies are needed to determine the coldest temperature where uptake is possible. In microbial ecology, the mechanisms of dispersal of extremophilic microorganisms such as nonsporulating obligate thermoacidophiles remain a long-standing mystery. Microorganisms that are taxonomically and genomically closely related are often found across geographically distant and disconnected thermal habitats[73,74]. Despite theory on a strong geographic barrier and endemism[75,76], pioneer populations must be seeded from somewhere. How could these habitat specialists survive across temperate environments to distant specialized habitats? Here we propose hidden metabolic flexibility enables cells to meet energy needs during dispersal. During the dispersal and transition throughout temperate environments, trace gas oxidizers such as *A. brierleyi* may rely on the continual acquisition of atmospheric $H_2$ and CO as universal metabolizable substrates for persistence. This is supported by our observation that trace $H_2$ consumption, including at atmospheric levels, enabled carbon-starved cells to maintain a higher ATP level at 25 °C. This may also help explain why terrestrial geothermal habitats select for diverse high-affinity trace gas oxidizers, including from the genera *Pyrinomonas*[6,44], *Chloroflexus*[16], *Kyrpidia*[46], *Thermomicrobium*, *Thermogemmatispora*[16,45], and *Methylacidiphilum*[47], despite being typically enriched with geothermally derived $H_2$.

Lastly, atmospheric $H_2$ oxidation may be a wider metabolic trait among aerobic archaea. The full set of four hydrogenases encoded by *A. brierleyi* are often possessed by other Sulfolobales members, including from the genera *Acidianus*, *Metallosphaera*, and *Sulfurisphaera* (Fig. 3). Certain halophilic (e.g. *Halopiger salifodinae*) and thermophilic (e.g. *S. islandicus* strain M.16.2) archaea encode [NiFe]-hydrogenases clustered with the high-affinity group 1h and 1l subgroups (Fig. 3). It is probable that at least some aerobic archaea consume atmospheric $H_2$ through these enzymes. However, it remains unresolved whether archaeal trace gas oxidizers make a quantitatively significant contribution to the overall biogeochemical cycle of atmospheric $H_2$ and CO, given they are vastly outnumbered by bacteria in the temperate soils that serve as the main sinks for these gases[17]. Altogether, these findings suggest that atmospheric $H_2$ oxidation is mediated by a broader range of microorganisms and enzymes than previously realized, and has potentially evolved at multiple occasions than initially hypothesized[77].

## Methods

### Archaeal strain and growth conditions

Pure cultures of the thermoacidophilic archaeal strains *Acidianus brierleyi* (DSM 1651[T]) and *Metallosphaera sedula* (DSM 5348[T]) were imported from Leibniz Institute DSMZ - German Collection of Microorganisms and Cell Cultures (DSMZ) in November 2021 and October 2022, respectively. *A. brierleyi* was regularly maintained in DSMZ medium 150, except that elemental sulfur was excluded. Per litre water, modified DSMZ medium 150 has a composition of 3 g $(NH_4)_2SO_4$, 0.5 g $K_2HPO_4 \cdot 3 H_2O$, 0.5 g $MgSO_4 \cdot 7 H_2O$, 0.1 g KCl, 0.01 g $Ca(NO_3)_2$, and 0.2 g yeast extract, with pH adjusted to 1.5 – 2.5 by 10N $H_2SO_4$ before autoclaving. *M. sedula* was regularly maintained heterotrophically in modified DSMZ medium 88. Per litre water, modified DSMZ medium 88 has a composition of 1.3 g $(NH_4)_2SO_4$, 0.28 g $KH_2PO_4$, 0.25 g $MgSO_4 \cdot 7 H_2O$, 0.07 g $CaCl_2 \cdot 2 H_2O$, 0.02 g $FeCl_3 \cdot 6 H_2O$, and 0.2 g yeast extract, with pH adjusted to 2 by 10N $H_2SO_4$ before autoclaving. Yeast extract stock solutions (10% w/v) were autoclaved separately before adding to the mineral medium. Liquid cultures of *A. brierleyi* (70 °C) and *M. sedula* (65 °C) were incubated with 100 rpm agitation in ambient air in the dark, unless otherwise specified. Cultures were propagated every 10–15 days with 1–10% v/v culture as inoculum in fresh medium pre-warmed at respective incubation temperatures.

### Growth characterization and biomass quantification

To characterize heterotrophic growth patterns of *A. brierleyi*, 1% v/v late exponential growing cultures were inoculated in 30 ml respective fresh medium in 125 ml aerated conical flasks. Triplicate cultures were then incubated at 70 °C with shaking at 100 rpm in the dark. Growth was monitored spectrophotometrically by measuring culture optical density at 600 nm ($OD_{600}$; Eppendorf BioSpectrometer® Basic) for up to six weeks. To quantify biomass of *A. brierleyi* for $H_2$ uptake kinetics and survival assays, total cell protein was measured using the bicinchoninic acid protein assay (Sigma-Aldrich) against bovine serum albumin standards. Cell lysates for the assay were prepared as follows: 2 ml cultures were centrifuged at $20200 \times g$ for 30 min. The cell pellet was washed with 1 ml DSMZ medium 150 mineral base and centrifuged for 10 min, twice, to remove free proteins from the medium. The washed cell pellet was then resuspended in 500 μl of Milli-Q water and lysed through probe sonication (output setting: 4; 10 s twice; Branson Digital Sonifier 450 Cell Disruptor). Subsequently, the cell lysate was cooled to room temperature and 100 μl was added to working reagents.

### Genome analysis

The closed genome and proteome annotations of *A. brierleyi* DSM 1651[T][78] were accessed through NCBI GenBank[79] under the assembly accession number GCA_003201835.2. Predicted protein coding sequences were further annotated against NCBI Conserved Domain Database (CDD) v3.19[80] using rpsblast (-evalue 0.01 -max_hsps 1 -max_target_seqs 5) in BLAST+ v2.9.0[81] and the Pfam protein family database v34.0[82] using PfamScan v1.6 (default setting)[83]. Protein subcellular localization and the presence of internal helices of the gene were predicted using PSORTb v3.0.3 (--archaea)[84]. Metabolic pathway analysis was performed using DRAM v1.2.4[85] with KEGG protein database (accessed 22 November 2021)[86]. Catalytic subunits of [NiFe] hydrogenases were identified and classified using HydDB[50]. The R package gggenes v0.4.1 (https://github.com/wilkox/gggenes) was used to construct gene arrangement diagrams. All sequences, annotations, and genetic arrangements are summarized in Supplementary Dataset 4.

### Identification and phylogenetic analysis of group 1 and 2 [NiFe]-hydrogenase

All large subunit amino acid reference sequences of [NiFe]-hydrogenases were downloaded from HydDB[50], aligned using Clustal Omega v1.2.4[87] and used to build a profile hidden Markov model (HMM) using HMMER v3.3.2[88]. The HMM profile was used to search against the open

reading frames of *A. brierleyi* genome and all representative genomes from the Genome Taxonomy Database (GTDB) release 202[51]. Each match shorter than 100,000 residues and with a bit score $\geq$ 34.5 was retained and then filtered by the presence of both conserved proximal and medial CxxC motifs required to ligate [NiFe]-hydrogenase $H_2$-binding metal centers[18,89]. The matches were also manually inspected through a combination of CDD annotations[80] and phylogenetic analysis. A multiple sequence alignment was performed on the retained archaeal and HydDB group 1 and 2 [NiFe]-hydrogenase sequences using MAFFT-L-INS-i v7.505[90], with selected group 3 and 4 [NiFe]-hydrogenases as outgroups. The best-fit substitution model (Q.pfam +R10) was determined using ModelFinder[91] implemented in IQ-TREE2 v2.2.0[92]. A maximum likelihood phylogenetic tree was constructed using IQ-TREE2 v2.2.0[92] with 1000 ultrafast bootstrap replicates[93].

## Computation modeling of hydrogenase complexes

[NiFe]-hydrogenase models were generated using AlphaFold multimer v2.1.1 implemented on the MASSIVE M3 computing cluster[58,94]. The amino acid sequences for the large and small subunits for each of the [NiFe]-hydrogenases identified in *A. brierleyi* (group 1g, group 2e, novel Sulfolobales clade 1 and 2) were modeled both alone and with the sequences of putative complex partners present in the hydrogenase gene cluster. Models produced were validated based on confidence scores (pLDDT), with only regions with a confidence score of > 85 utilized for analysis. Where complexes were predicted, subunit interfaces were inspected manually for surface complementarity and the absence of clashing atoms. Interfaces were also analyzed for stability using the program QT-PISA[95,96], with only interfaces predicted to be stably utilized for analysis. To assign cofactors to the AlphaFold model subunits, the closest homologous structural domain for each modeled protein were identified by searching the PDB database using NCBI BLAST with the amino acid sequence of the protein as the query or the DALI server using the AlphaFold model as the query[97,98]. The homologous structures were aligned with the AlphaFold models in Pymol, and cofactors were added in corresponding positions to that of the experimental structures if all conserved coordinating residues were present. Cofactor position was then manually adjusted to optimize coordination and to minimize clashes. Structures were used to model cofactors for the hydrogenases and PDB files of the hydrogenase models are summarized in Table S1 and Supplementary Dataset 6, respectively.

## Hydrogen and carbon monoxide consumption measurement

Gas chromatography assays were carried out to test for the ability of *A. brierleyi* to oxidize $H_2$ and CO at mid-exponential ($OD_{600}$: 0.04–0.08 pre-$OD_{max}$) and stationary growth phase ($OD_{600}$: 0.04–0.08 post-$OD_{max}$). 30 ml culture was transferred into a sterile 160-ml serum vial. Cultures were then allowed to adapt to the experimental conditions (70 °C and agitation speed of 100 rpm) for at least 2 h before sealing the vial with butyl rubber stoppers. Butyl rubber stoppers throughout all trace gas experiments were pre-treated with 0.1 N hot NaOH solution, according to the methods of Nauer et al.[48], to decrease abiotic emissions of $H_2$ and CO from the stopper. The ambient air headspace was amended with $H_2$, CO and $CH_4$ (via a mixed gas cylinder containing 0.1% v/v $H_2$, CO and $CH_4$ each in $N_2$, BOC Australia) to give initial mixing ratios of approximately 10 parts per million (ppmv) for each gas. $CH_4$ was included as an internal standard to check for any gas leakage. To monitor changes in headspace gas concentrations, 2 ml of headspace gas was withdrawn using a gas-tight syringe at each sampling time-point. A VICI gas chromatographic machine with a pulsed discharge helium ionization detector (model TGA-6791-W-4U-2, Valco Instruments Company Inc.) was then used to quantify $H_2$ and CO concentrations. The machine was calibrated against ultra-pure $H_2$ and CO standards down to the limit of quantification ($H_2$: 20 ppbv; CO: 9 ppbv)[45]. The machine was regularly calibrated against various calibration mixed gases with known trace gas concentrations. For *A. brierleyi*,

the assays were also carried out at incubation temperatures of 10 °C, 25 °C (room temperature), and 37 °C to determine thermal plasticity of trace gas oxidation activities. For all conditions, a heat-killed culture (autoclaved at 121 °C, 15 p.s.i. for 30 min) and a medium only control were included. The same experimental set up was also used to measure $H_2$ consumption by mid-exponentially growing *M. sedula* culture, except that the incubation temperature was 65 °C.

## *Acidianus brierleyi* hydrogen uptake kinetic analysis

Gas chromatography assays were used to determine uptake kinetics of $H_2$ by *A. brierleyi*. Quadruplicate cultures at exponential growth phase were incubated independently with a wide range of starting headspace $H_2$ concentrations (approximately 10, 20, 40, 80, 160, 320, 640, 1500, 3000, 6000, and 10000 ppmv) at 70 °C. Headspace $H_2$ concentrations from 10 to 1500 ppmv and from 3000 to 10000 ppmv were amended via 1% v/v $H_2$ (in $N_2$; Air Liquide) and ultra-high purity $H_2$ (99.999%; BOC Australia), respectively. Quadruplicate medium-only controls were also included for each concentration to monitor any gas leakage or sampling loss. Ten minutes prior to the first gas sampling, vial headspace over-pressure was briefly released by a 22G needle within the incubator at 70 °C to equilibrate with atmospheric pressure. Headspace $H_2$ concentrations were quantified at four time points, 0 h, 1 h, 2 h, and 4 h. To account for any potential lag response of the cultures to the concentrations of $H_2$ supplemented and allow sufficient time for equilibration of the headspace $H_2$ with the aqueous phase, reaction rates of $H_2$ consumption were calculated based on the concentration change between the third and fourth time points. The measurement of anaerobic uptake kinetics of $H_2$ by triplicate *A. brierleyi* was prepared using the exponential growth culture maintained with 0.3 g of tyndallized elemental sulfur powder as the electron acceptor in a $N_2$ headspace.

The dissolved concentrations of gases in aqueous phase at equilibrium state and at 1 atmospheric pressure were calculated according to Henry's law and van' t Hoff equation as follows:

$$k_T = k' \cdot e^{\frac{-\Delta_{soln}H}{R}\left(\frac{1}{T} - \frac{1}{298.15}\right)} \tag{1}$$

where $k_T$ and $k'$ denotes Henry's law constant of $H_2$ at the incubation temperature (i.e. 343.15 K) and at 298.15 K, respectively, $\Delta_{soln}H$ is the enthalpy of solution and $R$ is the ideal gas law constant. The constants $k'$ and $\frac{-\Delta_{soln}H}{R}$ were obtained from Sander[99]. Michaelis-Menten curves and parameters were estimated using the nonlinear fit (Michaelis-Menten, least squares regression) function in GraphPad Prism (version 9.3.1).

## Survival assays

The viability of *A. brierleyi* cultures was tested at two temperatures (25 °C and 70 °C) after supplementation of headspace $H_2$ at four low concentrations (0, 0.6, 5, and 50 ppmv). For each condition in triplicate, 40 ml of *A. brierleyi* cells at stationary phase (3 days post $OD_{max}$) was transferred to a sealed 120-ml serum vial with ambient air headspace. Four medium-only controls were included. Gas quantities equivalent of 0.6, 5, and 50 ppmv of headspace $H_2$ was supplemented to the vial daily using 1% v/v $H_2$ gas cylinder (in $N_2$; Air Liquide) whereas the culture without $H_2$ supplementation was added with 0.5 ml ultra-high purity $N_2$ (99.999%; BOC Australia) in the vial headspace daily. At each timepoint, 3 ml of culture was removed using a sterile 5-ml syringe fitted with a sterile needle. Cell viability was monitored by measuring three parameters: culture optical density at 600 nm; cell protein in culture through the BCA assay; and ATP in culture. A luciferin-luciferase-based assay (BacTiter-Glo, Promega) was used to measure culture ATP concentrations as previously described[100]. Briefly, 20 µl of culture was combined with 90 µl of BacTiter-Glo reagent in each well of a black 96-well plate (flat bottom, black polystyrene, Costar), followed by 5 min incubation at 25 °C and at 100 rpm in darkness. A multi-function plate reader (Infinite 200Pro, Tecan) was used to measure

luminescence with 1000 ms integration and automatic attenuation. Serially diluted ATP standards (Invitrogen) were included for each assay to calculate ATP concentrations in samples.

## Cellular fractionation

To separate cytosolic and membrane fractions of *A. brierleyi* cells, 4 L mid-exponential cultures were grown organoheterophically in either ambient air or 20% $H_2$ headspace. The cells were pelleted by centrifugation at 13,880 × $g$ for 10 min with a Sorvall High-Speed centrifuge. The supernatant was carefully poured off and the cell pellet was washed with approximately 100 ml of fractionation buffer (50 mM Tris, 200 mM NaCl, pH 7.00) and pelleted again by centrifugation at 13,880 × $g$ for 10 min. After discarding supernatant, the cell pellet was resuspended in 2 ml of fractionation buffer containing the following additives: 1× complete protease inhibitor (Roche), 0.1 mg DNase I dry powder, and $MgCl_2$ (1 mM final concentration). Resuspended cells were lysed using a cell disruptor (Constant Systems Ltd.) at a pressure of 40,000 psi, collected and centrifuged in a tabletop centrifuge at 2,400 × $g$ for 5 min to pellet any large cellular debris. A 100 μl aliquot of the supernatant containing the whole cell lysate was saved for downstream analysis. The remaining whole cell lysate was carefully pipetted into a clean round bottom ultracentrifuge tube (Beckman Coulter) and centrifuged at 132,617 × $g$ for 1 hr at 4 °C in a Sorvall WX Ultra 100 ultracentrifuge (Thermo Scientific) with a Type 70.1 Ti fixed angle rotor (Beckman Coulter). The resulting supernatant containing the cytosolic fraction was carefully decanted and saved for downstream analysis. The membrane fraction pellet was gently resuspended in 500 μl fractionation buffer with the addition of 3% (w/v) n-Dodecyl-β-D-maltopyranoside (Anatrace) and incubated at room temperature for 5 min with gentle nutation. The solubilized membrane fraction was centrifuged at 2,400 × $g$ for 5 min to pellet any insoluble debris and the resulting supernatant was saved for downstream analysis.

## BN-PAGE and hydrogenase activity staining

Samples were prepared for blue native polyacrylamide gel electrophoresis (BN-PAGE) with the addition of 5% (w/v) G-250 and 4× NativePAGE loading dye before being loaded onto a NativePAGE 4–16% Bis-Tris mini gel (Invitrogen). The Cathode buffer contained a final concentration of 5% (v/v) Cathode dye (Invitrogen) and 0.03% (w/v) n-Dodecyl-β-D-maltopyranoside (Anatrace) in a 1× Anode buffer solution (Invitrogen). Gels were run at a constant 150 V for 90 min at room temperature. On completion, gels were submerged in activity staining solution (50 μM nitrotetrazolium blue chloride (NBT), 50 mM Tris, 200 mM NaCl, pH 7.00) and placed in a gas-tight anaerobic jar (Oxoid Ltd.). The jar was purged of oxygen with a Büchi Vac V-500 vacuum and repressurized with a gas mix containing v/v 7% $H_2$, 7% $CO_2$, and 86% $N_2$. This process of vacuuming and repressuring with anaerobic gas was repeated six times to ensure an anaerobic environment. Gels were incubated at room temperature for 72 h before being visualized. Hydrogenase activity was inferred from the reduction of NBT, which forms a purple-colored precipitate when reduced. Gel bands displaying hydrogenase activity were excised and sent for hydrogenase identification through the proteomics pipeline described below.

## Quantitative proteomics

Shotgun proteomics was performed to characterize the cellular adaptation and metabolic remodeling of *A. brierleyi* at different growth stages and availability of growth substrates. Quadruplicate cultures were harvested at four various conditions: (a) mid-exponential growth phase on heterotrophic medium; (b) stationary phase on heterotrophic medium; (c) transition from heterotrophic to aerobic hydrogenotrophic growth on mineral medium; (d) transition from heterotrophic to sulfur-dependent anaerobic hydrogenotrophic

growth on mineral medium. Sixteen independent cultures for proteomics were first prepared by inoculating 1% v/v late exponential growing culture in 30 ml DSM medium 150 in 125 ml aerated conical flasks at incubation conditions previously described.

Cultures for conditions (a) and (b) were harvested when reaching $OD_{600}$ values 0.04–0.08 pre-$OD_{max}$ (mid-exponential growth phase) and $OD_{600}$ values 0.04–0.08 post-$OD_{max}$ (stationary growth phase), respectively. Cells maintained at autotrophic conditions (c) and (d) were prepared as follows: upon reaching a pre-$OD_{max}$ $OD_{600}$ of approximately 0.05, 15 ml culture was transferred into a 100kDa Amicron® Ultra-15 Centrifugal Filter Unit. The tube was centrifuged at 3000 × $g$ at room temperature for 5 min to separate living cells from the medium. The process was repeated once with the remaining culture. Cells on filter membrane were resuspended and washed by 3 ml of DSMZ medium 150 mineral base (without organic substrates) and the unit was centrifuged for 3 min to remove residual heterotrophic substrates. The wash step was repeated. Cells were then resuspended in 1 ml of DSMZ medium 150 mineral base and inoculated into 30 ml mineral base in a sterile 160-ml serum vial. The mineral base, per liter, was prior supplemented with 1 ml trace element solution containing (mg $l^{-1}$) 34.4 $MnSO_4$ • $H_2O$, 50.0 $H_3BO_3$, 70.0 $ZnCl_2$, 72.6 $Na_2MoO_4$ • 2 $H_2O$, 20.0 $CuCl_2$ • 2 $H_2O$, 24.0 $NiCl_2$ • 6 $H_2O$, 80.0 $CoCl_2$ • 6 $H_2O$, 1000 $FeSO_4$ • 7 $H_2O$, 3.0 $Na_2SeO_3$ • $5H_2O$, and 4.0 $Na_2WO_4$ • $2H_2O$ for autotrophic growth[101]. For condition (c), the sealed vial was flushed with 1 bar compressed air (Industrial grade; BOC Australia) fitted with a 0.22 μm syringe filter for 2 min to establish a stable aerobic condition. 20% $H_2$ and 5% $CO_2$ v/v in vial air headspace was prepared through ultra-high purity $H_2$ and $CO_2$ amendment. For condition (d), 0.3 g of tyndallized elemental sulfur powder was added to the culture and the vial was then sealed with a butyl rubber stopper. It was then flushed with 1 bar ultra-high purity $N_2$ (99.999%; BOC Australia) fitted with a 0.22 μm syringe filter for 2 min to establish an anaerobic condition and amended with a final $H_2$ and $CO_2$ concentration of 20% and 5%, v/v respectively. Cultures for both autotrophic conditions were allowed to grow at 70 °C and 100 rpm agitation for 48 hrs before harvesting.

To harvest cells for proteomic analysis, cell pellets from 20 ml cultures were collected by centrifugation (4800 x $g$, 50 min, −9 °C). Cell pellets were washed and resuspended with 2 ml DSMZ medium 150 mineral base in an Eppendorf tube, and centrifuged at 20200 × $g$ at −9 °C for 10 min. For cultures maintained at condition (d), elemental sulfur powder was initially precipitated by centrifuging at 1000 rpm at −9 °C for 1 min and the supernatant was transferred to a new Falcon tube before further centrifugation to collect cells. To further remove free proteins, cell pellet was washed and resuspended in phosphate-buffered saline (PBS; 137 mM NaCl, 2.7 mM KCl, 10 mM $Na_2HPO_4$ and 2 mM $KH_2PO_4$, pH 7.4), and centrifuged again, twice. Collected cell pellets were immediately stored at −20 °C and sent to the Proteomics & Metabolomics Facility in Monash University for analysis.

The samples were lysed in SDS lysis buffer (5% w/v sodium dodecyl sulfate, 100 mM HEPES, pH 8.1), heated at 95 °C for 10 min, and then probe-sonicated before measuring the protein concentration using a bicinchoninic acid (BCA) assay kit (Pierce). Equivalent amounts of lysed samples were denatured and alkylated by adding TCEP (tris(2-carboxyethyl) phosphine hydrochloride) and CAA (2-chloroacetamide) to a final concentration of 10 mM and 40 mM, respectively, and the mixture was incubated at 55 °C for 15 min. The reduced and alkylated proteins were then immobilized in S-Trap mini columns (Profiti) and sequencing grade trypsin was added at an enzyme to protein ratio of 1:50 and incubated overnight at 37 °C. Tryptic peptides were sequentially eluted from the columns using (i) 50 mM TEAB, (ii) 0.2% v/v formic acid and (iii) 50% v/v acetonitrile and 0.2% v/v formic acid. The fractions were pooled and concentrated in a vacuum concentrator prior to MS analysis.

The same amount of peptides was injected into the mass spectrometer for each sample to allow accurate quantification of protein

abundances and fair comparison across samples. For the label-free quantification (LFQ) analysis, we used a Dionex UltiMate 3000 RSLCnano system equipped with a Dionex UltiMate 3000 RS auto-sampler, an Acclaim PepMap RSLC analytical column (75 μm x 50 cm, nanoViper, C18, 2 μm, 100Å; Thermo Scientific) and an Acclaim PepMap 100 trap column (100 μm x 2 cm, nanoViper, C18, 5 μm, 100Å; Thermo Scientific), the tryptic peptides were separated by increasing concentrations of 80% acetonitrile (ACN) / 0.1% formic acid at a flow of 250 nl min$^{-1}$ for 158 min and analyzed with a QExactive HF mass spectrometer (ThermoFisher Scientific). The instrument was operated in data-dependent acquisition mode to automatically switch between full scan MS and MS/MS acquisition. Each survey full scan (375 – 1575 m/z) was acquired with a resolution of 120,000 (at 200 m/z), an AGC (automatic gain control) target of 3 x 10$^6$, and a maximum injection time of 54 ms. Dynamic exclusion was set to 15 s. The 12 most intense multiply charged ions ($z \geq 2$) were sequentially isolated and fragmented in the collision cell by higher-energy collisional dissociation (HCD) with a fixed injection time of 54 ms, 30,000 resolution and an AGC target of 2 x 10$^5$.

To obtain intensity-based absolute quantification (iBAQ) values of proteins, samples were analyzed on an Orbitrap Exploris 480 mass spectrometer (ThermoFisher Scientific) coupled to the same Nano LC system (Dionex Ultimate 3000 RSLCnano) as described above. The mass spectrometer was operated in data-dependent acquisition mode using in-house optimized parameters with 120 min of chromatographic separation used for each fraction. The acquisition used two FAIMS compensation voltages (−45, −75) operated under default standard resolution with an ion transfer tube temperature of 300 °C and a carrier gas flow rate of 4.6 L/min. Precursor ion scans were performed at a 60,000 resolution from 350 – 1,200 m/z, an AGC target of 300% and ion injection time set to auto. Fragmentation scans were performed at a resolution of 15,000 with normalized collision energy set to 28. Dynamic exclusion was applied for 45 s across all compensation voltages with only one charge state per precursor selected for fragmentation.

For the label-free quantification (LFQ) analysis, the raw data files were analyzed with the Fragpipe software suite v17.1 and its implemented MSFragger search engine v3.4[102] to obtain protein identifications and their respective label-free quantification (LFQ) values using standard parameters. Standard peptide modification was as follows: carbamidomethylation at cysteine residues was set as a fixed modification, as well as oxidation at methionine residues, and acetylation at protein N-terminal were set as variable modifications. For iBAQ analysis, protein identification and quantification was carried out with the MaxQuant software suite (version 2.2.0.0) utilizing standard parameters[103,104]. For all experiments, peptides and their corresponding proteins groups were both filtered to a 1% FDR with Percolator[105]. These data were further analyzed with LFQ-Analyst[106] as follows. First, contaminant proteins, reverse sequences and proteins identified "only by site" were filtered out. Also removed were proteins only identified by a single peptide and proteins that have not been identified consistently. The LFQ data was converted to log2 scale, samples were grouped by conditions and missing values were imputed using the 'Missing not At Random' (MNAR) method, which uses random draws from a left-shifted Gaussian distribution of 1.8 StDev (standard deviation) apart with a width of 0.3. Protein-wise linear models combined with empirical Bayes statistics were used for the differential expression analyses. The R package 'limma' was used to generate a list of differentially expressed proteins for each pair-wise comparison. A cutoff of the 'adjusted p-value' of 0.05 (Benjamini-Hochberg method) along with a |log2 fold change| of 1 was applied to determine significantly differentially abundant proteins in each pairwise comparison.

**Reporting summary**
Further information on research design is available in the Nature Portfolio Reporting Summary linked to this article.

## Data availability
The mass spectrometry proteomics data have been deposited to the ProteomeXchange Consortium via the PRIDE[107] partner repository with the dataset identifier PXD040286. All other study data are included in the article and/or supporting information. Source data are provided with this paper.

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

## Acknowledgements

The study was supported by an ARC Discovery Project Grant (DP200103074; to C.G. and R.G.), an NHMRC EL2 Fellowship (APP1178715; to C.G.), a Human Frontiers Science Program Grant (RGY0058/2022; to C.G.), a Monash International Tuition Scholarships (to P.M.L.), an Australian Government Research Training Scholarships (to P.M.L.), a Monash Graduate Research Completion Award (to P.M.L.), a Monash Postgraduate Publications Award (to P.M.L.), a Monash FMNHS Early Career Post-doctoral Fellowship (ECPF23-1113137961; to P.M.L.), and a Monash Summer Studentship (to E.T-M.). We thank Thanavit Jirapanjawat for technical support. This study used BPA-enabled (Bioplatforms Australia) / NCRIS-enabled (National Collaborative Research Infrastructure Strategy) infra-structure located at the Monash Proteomics and Metabolomics Facility and the MASSIVE M3 supercomputing infrastructure.

## Author contributions

C.G. and P.M.L. conceived, designed and supervised the study. Different authors were responsible for culture preparation and maintenance (P.M.L., E.T-M. and L.J.), growth analysis (P.M.L. and E.T-M.), genome analysis (P.M.L.), phylogenetic analysis (P.M.L., C.G., and M.M.), protein structural modeling (R.G., P.M.L., and C.G.), gas chromatography measurements (P.M.L., E.T-M., and L.J.), $H_2$ uptake kinetic characterization (P.M.L.), archaeal survival assay (P.M.L.), biochemical characterization (J.P.L., P.M.L., R.G., and A.K.), shotgun proteomics (P.M.L., H.L., I.H., E.T., and R.B.S.), and ecological theory (P.M.L., C.G., M.B.S., C.R.C. and H.A.P.). P.M.L., C.G., and R.G. analyzed data and wrote the manuscript with inputs from all authors.

## Competing interests

The authors declare no competing interests.
