## [Peer Review File · Nature Communications]

Trace gas oxidation sustains energy needs of a thermophilic archaeon at suboptimal temperaturesReviewer #1 (Remarks to the Author):

The paper by Lueng et al describes a careful set of experiments where archaea were grown in controlled conditions at very low (atmospheric) concentrations of H₂ and CO to assess whether they are capable of gaining energy by oxidizing these atmospheric trace gases. The work appears to be high quality, and I have no major comments on the quality of the data. The results appear to be quite solid, and their attention to scientific rigor is appreciated. My major criticism deals with, what in my opinion, would equate to an over stated importance of (1) the discovery of H₂ oxidation in aerobic archaea that was already known 30 years ago, and (2) that these archaea are an important sink of atmospheric trace gases.

As the authors point out, aerobic H₂ oxidation by archaea is not new and has been shown in numerous studies already going back 30 years. This is referenced in the authors manuscript in references 30-34. For example, going back several decades Setter et al was one of the first to show H₂ oxidation in thermophilic archaea. The authors comment that the enzymes involved in the process were not resolved, but it is safe to say that one could assume hydrogenase were required. Indeed, reference 35 as the authors point out has already shown that NiFe hydrogenase in *Metallosphaera sedula* showed that this is a hydrogenase that is very likely responsible for H₂ oxidation in archaea. The authors show there is a diversity of novel NiFe hydrogenases encoded by the archaea and do some nice predictive work with alpha fold and proteomics expression. This work is appreciated, but some of the statements made in the paper and conclusions are overstated in my opinion. For example, in the abstract it is written that "...it was not known whether archaea also use atmospheric H₂". I suppose the keyword here is 'atmospheric' since it has been known for decades that aerobic archaea use H₂ and oxidize it under aerobic conditions. And, that the authors main conclusion is that the archaea can use the trace H₂ in the atmosphere at ultra low concentrations. But, the importance of this seems to me a bit overstated since the thermophilic archaea live in geothermal environments that tend to have high H₂ concentrations and so the importance of atmospheric H₂ for these archaea in my opinion is likely to be insignificant. I am prepared to be wrong in this assessment but in order to test this possibility the authors would have to grow these archaea at high and low (atmospheric) H₂ concentrations, and at a range of temperatures representing different types of habitats (are the archaea oxidizing atmospheric H₂ at low temps outside of geothermal settings?) and see where they grow better. Do they grow better at low H₂, how does temperature affect atmospheric H₂ oxidation? These tests would also need to include controls to see whether these archaea more important than bacteria for atmospheric H₂ oxidation at low temperatures. That would be something that needs to be tested to draw conclusions about the importance of archaea as a global sink for atmospheric trace H₂ and CO (As stated in the last sentence of the abstract). But, such experiments were not done and so the importance of archaea as an important sink of atmospheric H₂ and CO remains unconstrained by the authors study (contrary to the authors conclusion).

Given that aerobic H₂ oxidation in archaea has been known for a very long time the novelty of the results are not great enough to justify publication in Nature Communications, in my opinion. That all being said, it is a solid piece of work with well designed experiments and would make a nice contribution to a more specialized microbiology journal as a deep dive into the genetic diversity and proteomics of novel archaean hydrogenases.

Reviewer #2 (Remarks to the Author):

Lueng et al discover that Archaea are capable of oxidizing trace H₂ even at concentrations as low as atmospheric levels, a feature hitherto only observed in bacteria. The discovery of this ability is of high value to the community and the methodology and execution are solid. Although biochemical demonstration of which hydrogenases are capable of oxidizing atmospheric levels of H₂ would of course be interesting, the focus here is the discovery of the ability itself so I believe that the study holds as it is. My only concerns are in how some of the data is interpreted. Including the title, my impression is that the authors are stretching the results more than it should

be, even though, in my eyes, the raw results and the more direct interpretations/conclusions presented are valuable alone.

Title:

This title makes it sound like a major portion of the domain archaea conserves energy by atmospheric trace gas oxidation, which is not true.

Electron bifurcation:

Why do the authors invoke electron bifurcation? To date, electron bifurcation is only known to be mediated by flavins (e.g., electron-bifurcating hydrogenase) or quinones and, for the latter, only cytochrome bc₁/b₆f have been reported to have quinone-mediated electron bifurcation activity.

Migration:

Though an interesting interpretation, the obtained results alone are insufficient to propose that oxidation of trace gases may allow for thermophiles to migrate through temperate habitats to distant high-temperature environments. If the authors would like to add data to support this, at a minimum, an experiment comparing the viability of *A. brierleyi* cells after incubation at lower temperatures over weeks with or without atmospheric concentrations of H₂ would be necessary.

Evolution:

The presented data does not provide any phylogenetic/evolutionary information to support "a deeper-rooted evolutionary origin" or "ancient origin" of atmospheric H₂ oxidation. We do not know how much of a biochemical hurdle there is for modification of an existing hydrogenase to interact with atmospheric levels of H₂. It could very well be that convergent evolution of such a capacity is common. Likewise, the term "cross-domain" is not appropriate as there is no evidence to say that there is a clear relationship between connecting the hydrogenases involved in the atmospheric H₂ consumption in bacteria and archaea. Moreover, such conclusions cannot be drawn without pinpointing which hydrogenase(s) facilitate oxidation of atmospheric levels of H₂. Similarly, the statement "SUL2 may thus represent a key 'missing link' to study the evolution of uptake hydrogenases" holds no meaning without phylogenetic analyses. If the authors would still like to discuss this in the manuscript, the authors ought to present phylogenetic evidence for ties between archaeal and bacterial oxidation of atmospheric levels of H₂.

- Masaru K. Nobu

Reviewer #3 (Remarks to the Author):

This study by Leung et al. provides compelling evidence that the oxidation of trace-gases to sub-atmospheric levels is possible by thermophilic aerobic archaea. Through high-sensitivity gas chromatography they show the consumption of both H₂ and CO by *Acidianus brierleyi* from levels 15-fold higher than atmospheric down to below 40% the atmospheric concentration. Remarkably, H₂ and CO were oxidized by *A. brierleyi* at temperatures below 37C, far cooler than the 70C optimal growth temperature of this organism. This process was also shown for a different archaeon, *Metallosphaera sedula*, which consumed even more of the H₂. To understand how *A. brierleyi* did this, the genome was searched for homologs of hydrogenases which were then subjected to phylogenetic and modeling analyses to infer function. Four distinct [NiFe] uptake hydrogenases were identified, two had similarity to hydrogenases previously suggested to mediate H₂ oxidation from other Sulfolobales, while the other two hydrogenases belong to unique subgroups that are found in other Sulfolobales genomes but are uncharacterized. Differential proteomic analysis was used to determine how protein expression profiles varied between cells grown heterotrophically during log or stationary phase versus autotrophic growth under anaerobic or aerobic conditions. The proteomic results revealed clear differences in expression of proteins related to cellular metabolism that were consistent with expectations for each growth condition, and allowed the authors to determine that group 1g [NiFe] hydrogenases followed by the novel SUL1 group were most up-regulated during anaerobic growth on sulfur, while the novel SUL2 was

specifically up-expressed under aerobic autotrophic conditions. The fourth hydrogenase, belonging to group 2e, was found in low abundance in all four conditions. These results suggest that the H₂ scavenging ability of *A. brierleyi* for the purposes of energy conservation is mediated by distinct hydrogenases depending on growth conditions, while group 2e expression is mostly constitutive. The authors end the paper with a fascinating discussion about how atmospheric H₂ oxidation by thermophiles at low temperatures may be related to their survival during long-range dispersal—an important question yet to be resolved.

For figure 1, where is the control data where no excess H₂ was provided? Part of conserving energy would be demonstration of growth due to the process.

Fig 3. Many different font sizes make it difficult to really appreciate details. Must zoom considerably to see legend for 3A.

General comments:

I would like to have seen H₂ uptake kinetics for *A. brierleyi* grown under anaerobic conditions with S₀ using atmospheric levels of H₂. These are important data to be considered in light of the proteomics results and to better understand the limits of this organism's ability to scavenge H₂.

Related to the above comment, I think the title is perhaps too broad. You need stronger data to show that energy is conserved versus a control. Also, you focus only on aerobic trace gas oxidation. So that should be reflected in the title, unless more comprehensive anaerobic data are also included.

Further experiments are defined to test the kinetics of purified proteins, or to perform heterologous expression in other Sulfolobales. I agree these are good suggestions and think it would also be interesting to investigate the effect of mutating these genes and testing the effect on atmospheric H₂ oxidation in whole cells. If this was done, and cells did not grow with trace H₂, it would bolster the argument they are actually conserving energy.

In the methods it says "organic substrate stock solutions. . ." was this really only yeast extract? Or were other organics used that were not specified in the methods? It should also be made clearer this was not added to autotrophic conditions.

In the materials and methods it mentions that biomass was quantified by OD as well as protein quantification. However, I do not see the protein biomass data presented in the figures in that way. It appears that these data were used for Fig 1C, so perhaps it makes more sense to talk about the protein quantitation in the kinetics section of the methods.

Reviewer #4 (Remarks to the Author):

Summary: The genome sequences of the two archaea of interest here clearly indicate the presence of the enzyme types of interest here. While this may have been reported as such in the literature, this comes as no surprise. This report shows the oxidation of low levels of hydrogen and carbon monoxide by the thermoacidophilic archaea *Acidianus brierleyi* and (briefly discussed) *Metallosphaera sedula*. It proposes that the putative genes encoding hydrogenases in *Acidianus brierleyi*, in addition to those already known in the Order Sulfolobales, are responsible for the observed phenomenon. Quantitative proteomics is used to assess the role of these putative hydrogenases in hydrogen oxidation during various modes of growth in *A. brierleyi*. It is pointed out that oxidation of H₂ and CO at trace atmospheric concentrations has only been shown in bacteria previously, and that the archaea also contribute to this cycle.

While this is an interesting observation, it does not 'move the needle' much. Yes, these archaea contribute to the cycling of CO and H₂ but at very low levels and mostly in thermal environments.

At lower temperatures, the metabolic rates of these archaea are vanishingly small. Since thermal environments make up a very small footprint on earth, the phenomenon described may not matter much in the overall scheme of things.

Comments:

- *Metallosphaera sedula* demonstrates similar trace gas oxidation capabilities to *Acidianus brierleyi*. While *M. sedula* consumes H₂ at a slower rate, it continues to consume H₂ at sub-atmospheric levels while *A. brierleyi* levels off. The authors then show that *M. sedula* and *A. brierleyi* have different types of hydrogenases. In identifying which hydrogenase is primarily responsible for trace gas oxidation in aerobic conditions, it would be helpful to provide proteomic analysis of *M. sedula* for comparison against *A. brierleyi*. Difference in hydrogenase levels/activities could help explain why *A. brierleyi* levels off at low H₂ concentrations while *M. sedula* continues to oxidize H₂.
- The claim that the hydrogenases are 'novel' needs to be supported better. It seems that these fall into the types of these enzymes that have been studied for decades. The 'novelty' may only come into play in terms of thermostability which can be related to the 'non-catalytic' amino acid sequence not involved in the active site.
- The authors frequently describe the hydrogenases as "constitutively expressed", while also showing varying levels of the hydrogenases depending on the mode of growth, implying the hydrogenase expression is highly regulated. These points are contradictory.
- The authors claim that trace gas oxidation could be providing maintenance energy for cells in a low metabolic state. It would be helpful to demonstrate that energy conservation is actually occurring to a biologically relevant extent at atmospheric levels of H₂, either by its effect on cell density or by reduction of an energy carrier through the electron transport chain.
- The structural comparison of the hydrogenases is all computational, making it very difficult to make claims about different types of electron acceptors or the comparative kinetic properties of the hydrogenases. Biochemical characterization of these different hydrogenases from *A. brierleyi* would add substantial support to the authors' claims about the various roles of each hydrogenase. As it stands, the Michaelis-Menten kinetic analysis only evaluates the whole cell behavior and does not provide insight into the differences in substrate affinity between the various hydrogenases.

Overall, the authors make some insightful claims, but further experimental support would be necessary to solidify the claims about certain hydrogenases being responsible for certain modes of growth or for the trace gas oxidation being able to provide sufficient maintenance energy to the cells. Also, they should do a calculation that estimates the contribution of these extreme thermophiles to the overall H₂ and CO cycling on this planet.

Summary of response to reviewers

We thank the editor for their careful consideration of this manuscript and the four reviewers for their thorough and constructive feedback. While all four reviewers agreed that our work is of high quality that advances the understanding of hydrogen (H₂) metabolism by aerobic archaea, they somewhat diverged in their opinions regarding the study's novelty and the direction of the revision. We have carefully revised our manuscript to address their concerns and incorporate their suggestions.

As detailed below, we have performed additional experiments and analyses to provide strong evidence that members of the archaea conserve energy by atmospheric trace gas oxidation and this process promotes long term survival. Considering the diverse nature of new experiments (gas chromatography, biochemical characterization, mass spectrometry, and survival assays), and the low growth yield and specific requirements of the archaeal strain, the extended time for revision was necessary. The new results necessitate the addition of two new authors (James Lingford and Ashleigh Kropp for biochemical characterization). We believe these substantial revisions have greatly improved the study such that it is worthy of publication in Nature Communications.

The major revisions include:

1. **New gas chromatography experiments** showing archaeal consumption of atmospheric trace gases occurs at ambient levels, and isn't an artifact stimulated by elevated concentrations, and a new kinetic analysis showing *Acidianus brierleyi* consumes H₂ at faster rates under aerobic conditions than sulfur-dependent growth (addressing comments from reviewer 3). This is provided in the **new Fig. 1D, Fig. S1, and Table S1**.
2. **Addition of viability assays** comparing cellular ATP, culture density, and cell protein on carbon-starved stationary phase *A. brierleyi* cells supplemented daily with trace H₂ equivalent to 0, 0.6, 5, and 50 ppmv at both 70°C and 25°C for 20 days. The results show that H₂ supplementation significantly enhanced cellular ATP, culture density, and cell protein in a dose-dependent manner, with a more pronounced effect for persisting cultures at 25°C compared to 70°C (**new Fig. 2D-E, Fig. S3; Table S2**). Specifically, cultures supplemented with atmospheric H₂ had 38% higher cellular ATP at day 20 than cultures without headspace H₂. The new result demonstrates the physiological relevance of atmospheric trace gas oxidation for energy conservation and survival of these archaea. Importantly, this provides strong support to our novel hypothesis that atmospheric trace gas oxidation is a dispersal trait promoting survival and potentially dispersal of obligate thermophiles across temperate environments. This addresses comments by all four reviewers.
3. **Addition of biochemical characterizations** of hydrogenases harvested from membrane and cytosolic fractions of *A. brierleyi* by activity staining and mass spectrometric identification. These data definitely show that the novel SUL2 hydrogenases are the active and dominant enzymes for aerobic H₂ consumption, while the other hydrogenases are minimally present under aerobic conditions. This experiment also provided further insights into the functions (electrogenic H₂ oxidation), complex formation (likely a four-subunit complex), and localization (preferentially membrane-associated) of this enzyme. The biochemical characterization addresses comments by reviewers 3 and 4. The new result is provided in the **new Fig. 5, Fig. S6, and Table S4**.
4. **Revision of the title** to address reviewers 2 and 3.

5. **Text revision to discuss new results** and address comments of reviewers 1 and 4 on the significance of archaeal trace gas oxidizers as biogeochemical sinks and study novelty

With the latest results and revision, we believe our study delivers high quality and novel results worthy for publication in Nature Communications. They include:

- *The discovery of the first atmospheric H₂ oxidizers in the domain archaea, overturning the previous paradigm that this metabolism is uniquely to bacteria*
- *The discovery of a second archaeal phylum (Thermoproteota) capable of consuming atmospheric CO*
- *The discovery of novel hydrogenases responsible for aerobic H₂ oxidation in archaea and resolving physiological roles of the multiple hydrogenases*
- *The first demonstration that atmospheric trace gas oxidation promotes long-term survival and is constitutively active beyond growth temperatures of extremophile archaea*
- *A new theory that energy harvesting from ubiquitous atmospheric trace gases is a trait that supports the dispersal of niche restricted microbes*

Responses to Reviewer #1

The paper by Leung et al describes a careful set of experiments where archaea were grown in controlled conditions at very low (atmospheric) concentrations of H₂ and CO to assess whether they are capable of gaining energy by oxidizing these atmospheric trace gases. The work appears to be high quality, and I have no major comments on the quality of the data. The results appear to be quite solid, and their attention to scientific rigor is appreciated. My major criticism deals with, what in my opinion, would equate to an over stated importance of (1) the discovery of H₂ oxidation in aerobic archaea that was already known 30 years ago, and (2) that these archaea are an important sink of atmospheric trace gases.

We thank the reviewer for their recognition of the quality of our study. We agree with the reviewer that aerobic H₂ oxidation by archaea has been recognised for a number of decades. However, prior to this study it was not known that archaea were capable of H₂ oxidation at atmospheric concentrations. This discovery has important ramifications for our understanding of how these microbes are able to persist when they are deprived of other energy sources, for example during dispersal. While the abundance of archaea in most aerobic environments is low compared to bacteria, understanding they are capable of atmospheric trace gas scavenging is important for fully understanding this process.

As the authors point out, aerobic H₂ oxidation by archaea is not new and has been shown in numerous studies already going back 30 years. This is referenced in the authors manuscript in references 30-34. For example, going back several decades Setter et al was one of the first to show H₂ oxidation in thermophilic archaea. The authors comment that the enzymes involved in the process were not resolved, but it is safe to say that one could assume hydrogenase were required. Indeed, reference 35 as the authors point out has already shown that NiFe hydrogenase in *Metallosphaera sedula* showed that this is a hydrogenase that is very likely responsible for H₂ oxidation in archaea. The authors show there is a diversity of novel NiFe hydrogenases encoded by the archaea and do some nice predictive work with alpha fold and proteomics expression. This work is appreciated, but some of the statements made in the paper and conclusions are overstated in my opinion. For example, in the abstract it is written that "...it was not known whether archaea also use atmospheric H₂". I suppose the keyword here is 'atmospheric' since it has been known for decades that aerobic archaea use

H₂ and oxidize it under aerobic conditions. And, that the authors main conclusion is that the archaea can use the trace H₂ in the atmosphere at ultra low concentrations. But, the importance of this seems to me a bit overstated since the thermophilic archaea live in geothermal environments that tend to have high H₂ concentrations and so the importance of atmospheric H₂ for these archaea in my opinion is likely to be insignificant. I am prepared to be wrong in this assessment but in order to test this possibility the authors would have to grow these archaea at high and low (atmospheric) H₂ concentrations, and at a range of temperatures representing different types of habitats (are the archaea oxidizing atmospheric H₂ at low temps outside of geothermal settings?) and see where they grow better. Do they grow better at low H₂, how does temperature affect atmospheric H₂ oxidation? These tests would also need to include controls to see whether these archaea more important than bacteria for atmospheric H₂ oxidation at low temperatures. That would be something that needs to be tested to draw conclusions about the importance of archaea as a global sink for atmospheric trace H₂ and CO (As stated in the last sentence of the abstract). But, such experiments were not done and so the importance of archaea as an important sink of atmospheric H₂ and CO remains unconstrained by the authors study (contrary to the authors conclusion).

*To further investigate the relevance of H₂ at nanomolar levels, including atmospheric concentrations, we have now performed viability assays on carbon-starved stationary phase *A. brierleyi* cells supplemented daily with trace H₂ equivalent to 0, 0.6, 5, and 50 ppmv at both 70°C and 25°C for 20 days. The results show that H₂ supplementation significantly enhances cellular ATP, culture density, and cell protein for cultures persisting at 25°C (new Fig. 2D-E; Table S2). Specifically, cultures supplemented with atmospheric H₂ had 38% higher cellular ATP at day 20 than cultures without headspace H₂. Conversely, H₂ has a comparatively small effect on the cellular viability at 70°C. The greater importance of trace gas oxidation at low temperatures for this obligate thermophile provides a strong support for our hypothesis on microbial biogeography that this trait is relevant for promoting their survival and dispersal across temperate environments not suited to *A. brierleyi* growth.*

We have now included the new result in L160 – 179 and expanded the discussion in L376 – 379 and L455 – 457.

L160 – 179: “We investigated if the metabolism of nanomolar H₂ enhances survival of *A. brierleyi* during carbon starvation. The viability of stationary phase cultures supplemented daily with a headspace of 0, 0.6 (ambient level in air), 5, and 50 ppmv H₂ was monitored during long-term incubations at both 70°C and 25°C. Optical density and cell protein declined more rapidly in persisting cultures at 70°C than at 25°C, reflecting increased cell lysis and a substantial energetic cost to maintain cell integrity in hot acidic conditions (Fig. S3). ATP levels, culture density, and cell protein concentrations varied between the incubations in a manner that reflects H₂ availability (Fig. S3). Effects were modest during incubations at 70°C: after 20 days of persistence, cellular ATP (nmol mg_{protein}⁻¹) of cultures supplemented with 50 ppmv H₂ was significantly greater than without H₂ by 27% at 70°C ($p < 0.05$) (Fig. 2D); there was no significant difference for 0.6 and 5 ppmv treatment compared to zero H₂ control, though ATP levels, optical density, and cell protein are generally higher (Fig. S3). H₂ supplementation at 25°C greatly enhanced cell viability. Cellular ATP was significantly greater across all H₂ treatments, including at atmospheric level, after 10 days of persistence (Fig. 2E); at day 20, cultures supplemented with 0.6, 5, and 50 ppmv H₂ had 38%, 129%, and 672% greater cellular ATP than the control, and were greater compared to time zero for cultures provided with 50 ppmv H₂ (Table S2). Concordant but weaker patterns were observed for culture density and cell protein measurements (Fig. S3). These results strongly suggest *A. brierleyi* conserves

energy from trace H_2 at nanomolar ranges during persistence and atmospheric substrates are significant for cells to stay energized at temperate conditions.”

Regarding the biogeochemical implication of archaeal trace gas oxidation, we did not claim archaea are more important than bacteria for atmospheric H_2 oxidation at low temperatures and nor did we make any statement about the relative significance of this domain in gas cycling. Instead, in our original abstract and conclusion, we made a factual statement based on our result that members of archaea are previously overlooked sink for atmospheric H_2 and CO (that this metabolism is not unique to bacteria). However, we agree with the reviewer that it isn't clear whether this contribution is quantitatively significant for the global cycling of these gases. Thus, to avoid confusion, we have replaced “sink” with “mediators”. In addition, we further clarified that the importance of archaea in biogeochemical cycling of H_2 and CO will require further analysis, especially ecosystem-scale samplings in **L467 - 471**.

L467 - 471: “However, it remains unresolved whether archaeal trace gas oxidisers make a quantitatively significant contribution to the overall biogeochemical cycle of atmospheric H_2 and CO, given they are vastly outnumbered by bacteria in the temperate soils that serve as the main sinks for these gases ¹⁷.”

Given that aerobic H_2 oxidation in archaea has been known for a very long time the novelty of the results are not great enough to justify publication in Nature Communications, in my opinion. That all being said, it is a solid piece of work with well designed experiments and would make a nice contribution to a more specialized microbiology journal as a deep dive into the genetic diversity and proteomics of novel archaean hydrogenases.

As we state in the manuscript, the novelty of this aspect of our work is not the observation that archaea are capable of aerobic H_2 oxidation, but the fact that they are capable of and benefit from oxidising H_2 at atmospheric concentrations. This process is incredibly important for bacterial persistence, but it has not been previously demonstrated for archaea. In addition to this major finding, this study provides a deeper mechanistic understanding of a known process (aerobic H_2 oxidation in archaea), but also provides a whole new lens on archaeal physiology by revealing that some members of this domain are capable of scavenging trace gases and this supports survival at a range of temperatures. Despite knowing that archaea are capable of aerobic H_2 oxidation, our understanding of the biochemistry and physiology that underpins this process has remained superficial. For example, the enzymatic determinants (e.g. which specific hydrogenases), properties (e.g. kinetics, consumption thresholds, temperature dependency), and physiological roles beyond growth (e.g. persistence) have not been demonstrated. Our work provides new insights and evidence of these unresolved questions in archaeal aerobic H_2 metabolism.

We make the following discoveries and insights:

Archaeal physiology and metabolism. We presented the first measurements and kinetic characterization that aerobic archaea represented by *A. brierleyi* and *M. sedula* aerobically and constitutively oxidize H_2 to nanomolar ranges, including at sub-atmospheric levels. Another novel realization is that the obligate thermophile can maintain trace gas metabolism at temperate temperatures far below their growth ranges. They conserve energy from nanomolar levels of H_2 to significantly enhance persistence during carbon starvation, with an increased importance at low temperatures. Through comparative proteomics, we further clarified the potential roles of the four hydrogenases in *A. brierleyi* under different redox and growth conditions.

Novel enzymatic determinants. We identified the hydrogenase determinants controlling *Sulfolobales* H₂ metabolism, including the findings of two novel major lineages (SUL1 and SUL2) through holistic phylogenetic, synteny, zymographic, structural modelling and proteomic analyses. The substantial evidence suggests that the novel SUL2 hydrogenase is actually determinant for archaeal aerobic H₂ oxidation, rather than the group 1g hydrogenase as previously assumed. Another novel finding is that the hydrogenases responsible for H₂ uptake in archaea are distinct from known high-affinity bacterial enzymes, revealing atmospheric H₂ oxidation had evolved independently and diverged early in evolution.

New mediators of atmospheric H₂ and CO biogeochemical cycling. With the above results, we demonstrated for the first time that atmospheric H₂ uptake is not unique to bacterial metabolism but members of archaea are also mediators, in addition to the first evidence that *Thermoproteota* species can consume atmospheric CO. Our report has major implications in the study of the origin of this process and provides a foundational work to study the importance and contribution of archaea in the biogeochemical cycling of H₂ and CO within geothermal habitats and beyond. The importance of atmospheric H₂ oxidation to the microbial ecology field is clearly exemplified by numerous high-profile publications of this finding in bacteria, e.g. *PNAS* 2014, 2015 (doi:10.1073/pnas.1407034111; doi:10.1073/pnas.1508385112), *Nature* 2017 (doi:10.1038/nature25014), *Nature Microbiology* 2021 (doi:10.1038/s41564-020-00811-w), *Nature* 2023 (doi:10.1038/s41586-023-05781-7). Similarly, the report of the first archaeal atmospheric CO oxidizer was premiered in *PNAS* 2015 (doi:10.1073/pnas.142498911).

New theory in microbial biogeography. We showed that survival of obligate thermophiles at temperate temperatures is enhanced by the consumption of trace gases, reconciling the widespread presence of atmospheric trace gas oxidizers isolated from geothermal habitats. This suggests that atmospheric trace gas oxidation may serve a further ecologically important role in supporting microbial survival and thus dispersal from their native geothermal habitats, providing a new energetics-based theory on a key unresolved question on microbial biogeography.

Responses to Reviewer #2

Leung et al discover that Archaea are capable of oxidizing trace H₂ even at concentrations as low as atmospheric levels, a feature hitherto only observed in bacteria. The discovery of this ability is of high value to the community and the methodology and execution are solid. Although biochemical demonstration of which hydrogenases are capable of oxidizing atmospheric levels of H₂ would of course be interesting, the focus here is the discovery of the ability itself so I believe that the study holds as it is. My only concerns are in how some of the data is interpreted. Including the title, my impression is that the authors are stretching the results more than it should be, even though, in my eyes, the raw results and the more direct interpretations/conclusions presented are valuable alone.

We are grateful for the reviewer's positive comments on the quality of our findings and his constructive feedback. As responded in detail below, we have included new experiments showing trace H₂ promotes viability of carbon-starved cells particularly at temperate conditions and have modified our result interpretation throughout to be more conservative.

Title:

This title makes it sound like a major portion of the domain archaea conserves energy by atmospheric trace gas oxidation, which is not true.

We agree the title may be misinterpreted as most archaea can conserve energy by atmospheric trace gas oxidation. We have revised the title more specifically to “Trace gas oxidation sustains energy needs of a thermophilic archaeon at suboptimal temperatures”.

Electron bifurcation:

Why do the authors invoke electron bifurcation? To date, electron bifurcation is only known to be mediated by flavins (e.g., electron-bifurcating hydrogenase) or quinones and, for the latter, only cytochrome bc1/b6f have been reported to have quinone-mediated electron bifurcation activity.

We included electron bifurcation as one of the possibilities explaining the structural models of the group 1g and SUL2 hydrogenase complexes. Our structural models predict that the four-subunit complexes of these two hydrogenases share a highly homologous structure (despite distant phylogenetic relationships). The complexes appear to have two alternative paths for the two electrons from the [NiFe] center to be transferred to quinone at Heme 2 site of Lsp1 subunit and potentially to an unknown electron acceptor at iron-sulfur cluster of Lsp2 subunit (Fig. 4). In order to maximise energy yield for ATP generation and provide powerful reductants for carbon fixation, we hypothesize that electrons from H₂ are transferred to high-potential quinone or a low-potential electron acceptor (e.g. ferredoxin) controlled by redox potentials (thermodynamics), or alternatively to both substrates via bifurcation. This efficient coupling may explain why the group 1g and SUL2 hydrogenases are dominantly expressed under anaerobic and aerobic hydrogenotrophic growth, respectively. We have cautioned that experiment validation will be necessary and discussed different possibilities of electron fate in both the results and discussion sections:

L405-413: *“Alternatively, as inferred from the AlphaFold2 structural model (Fig. 4), the SUL2 and group 1g enzymes may form electron-bifurcating complexes that relay electrons from H₂ to high-potential quinones through the Lsp1 subunit for proton-motive force generation and low-potential acceptors (e.g. ferredoxin) through the Lsp2 subunit for carbon fixation. Such an efficient energy conservation mechanism may explain the dominance of these hydrogenases during hydrogenotrophic growth. However, direct experimental validation using purified enzymes would be necessary to confirm this. It also cannot be ruled out that these complexes may alternate in the electron acceptors they couple to, for example in response to changes in cofactor ratios or through modularity in their structures.”*

Migration:

Though an interesting interpretation, the obtained results alone are insufficient to propose that oxidation of trace gases may allow for thermophiles to migrate through temperate habitats to distant high-temperature environments. If the authors would like to add data to support this, at a minimum, an experiment comparing the viability of *A. brierleyi* cells after incubation at lower temperatures over weeks with or without atmospheric concentrations of H₂ would be necessary.

*We thank the reviewer for the excellent suggestion. We have now performed viability assays on carbon-starved stationary phase *A. brierleyi* cells supplemented daily with trace H₂*

equivalent to 0, 0.6, 5, and 50 ppmv at both 70°C and 25°C for 20 days. The results show that H₂ supplementation, including at atmospheric concentrations, significantly enhanced cellular ATP, culture density, and cell protein for cultures persisting at 25°C (**Fig. 2D-E; Table S2**). This provides promising evidence that oxidation of the universal trace gases supports energy conservation and potentially the dispersal of aerobic thermophiles through temperate environments.

We have now included the new result in **L160 – 179** and expanded the discussion in **L376 – 379** and **L455 – 457**.

Evolution:

The presented data does not provide any phylogenetic/evolutionary information to support “a deeper-rooted evolutionary origin” or “ancient origin” of atmospheric H₂ oxidation. We do not know how much of a biochemical hurdle there is for modification of an existing hydrogenase to interact with atmospheric levels of H₂. It could very well be that convergent evolution of such a capacity is common.

Our phylogenetic analysis of the uptake [NiFe]-hydrogenases suggest the two hydrogenases (group 2e and novel SUL2) implicated in aerobic H₂ oxidation (and atmospheric H₂ oxidation) in Sulfolobales form distant clades from known high-affinity enzymes in bacteria. We agree with the reviewer’s critical comments that this alone is insufficient to differentiate if this early divergence indicates deep evolutionary origin or convergent evolution of atmospheric H₂ uptake hydrogenases. However, the analysis remains informative that hydrogenases mediating atmospheric H₂ has a wider distribution than previous known and that this trait is likely evolved at multiple occasions in the light of divergence of known high-affinity hydrogenases (group 1f, 1h, 1l, 2a, and SUL2/group2e).

We have thus refined and restricted our interpretation of phylogenetic analysis.

L471 – 473: “Altogether, these findings suggest that atmospheric H₂ oxidation is mediated by a broader range of microorganisms and enzymes than previously realised, and has potentially evolved at multiple occasions than initially hypothesised ⁷⁵.”

Likewise, the term “cross-domain” is not appropriate as there is no evidence to say that there is a clear relationship between connecting the hydrogenases involved in the atmospheric H₂ consumption in bacteria and archaea. Moreover, such conclusions cannot be drawn without pinpointing which hydrogenase(s) facilitate oxidation of atmospheric levels of H₂.

The term “cross-domain” was not intended to infer the transfer or connection between hydrogenases involved in the atmospheric H₂ consumption in bacteria and archaea. Rather, we intended it to describe that the phenomenon of atmospheric H₂ oxidation is a trait present in both archaea and bacteria (cross-domain). Given the ambiguity, we have removed ‘cross-domain’ in both instances it was used:

L43-46: “These findings also demonstrate that atmospheric H₂ consumption extends to the domain archaea and identifies previously unknown microbial and enzymatic mediators of atmospheric H₂ and CO consumption.”

L185-186: “Together, these results provide the first evidence that atmospheric H₂ oxidation is not exclusive to bacteria, but rather also extends to the archaeal domain.”

Similarly, the statement “SUL2 may thus represent a key ‘missing link’ to study the evolution of uptake hydrogenases” holds no meaning without phylogenetic analyses. If the authors would still like to discuss this in the manuscript, the authors ought to present phylogenetic evidence for ties between archaeal and bacterial oxidation of atmospheric levels of H₂.

We have now removed the statement that “SUL2 may thus represent a key ‘missing link’ to study the evolution of uptake hydrogenases”.

- Masaru K. Nobu

Responses to Reviewer #3:

This study by Leung et al. provides compelling evidence that the oxidation of trace-gases to sub-atmospheric levels is possible by thermophilic aerobic archaea. Through high-sensitivity gas chromatography they show the consumption of both H₂ and CO by *Acidianus brierleyi* from levels 15-fold higher than atmospheric down to below 40% the atmospheric concentration. Remarkably, H₂ and CO were oxidized by *A. brierleyi* at temperatures below 37C, far cooler than the 70C optimal growth temperature of this organism. This process was also shown for a different archaeon, *Metallosphaera sedula*, which consumed even more of the H₂. To understand how *A. brierleyi* did this, the genome was searched for homologs of hydrogenases which were then subjected to phylogenetic and modeling analyses to infer function. Four distinct [NiFe] uptake hydrogenases were identified, two had similarity to hydrogenases previously suggested to mediate H₂ oxidation from other Sulfolobales, while the other two hydrogenases belong to unique subgroups that are found in other Sulfolobales genomes but are uncharacterized. Differential proteomic analysis was used to determine how protein expression profiles varied between cells grown heterotrophically during log or stationary phase versus autotrophic growth under anaerobic or aerobic conditions. The proteomic results revealed clear differences in expression of proteins related to cellular metabolism that were consistent with expectations for each growth condition, and allowed the authors to determine that group 1g [NiFe] hydrogenases followed by the novel SUL1 group were most up-regulated during anaerobic growth on sulfur, while the novel SUL2 was specifically up-expressed under aerobic autotrophic conditions. The fourth hydrogenase, belonging to group 2e, was found in low abundance in all four conditions. These results suggest that the H₂ scavenging ability of *A. brierleyi* for the purposes of energy conservation is mediated by distinct hydrogenases depending on growth conditions, while group 2e expression is mostly constitutive. The authors end the paper with a fascinating discussion about how atmospheric H₂ oxidation by thermophiles at low temperatures may be related to their survival during long-range dispersal—an important question yet to be resolved.

We thank the reviewer for their thorough evaluation and recognition of the quality of our manuscript.

For figure 1, where is the control data where no excess H₂ was provided? Part of conserving energy would be demonstration of growth due to the process.

*We have now included a gas chromatography experiment showing *Acidianus brierleyi* rapidly consumes ambient level of H₂ (average 0.62 ppmv in the lab air) to sub-atmospheric levels at*

both 70°C and 25°C (**Fig. S1**). This shows that atmospheric H₂ uptake is not simulated by elevated H₂ but rather is a constitutive process. It should be noted that energy conservation can be used to support both survival and growth, with trace gas oxidation primarily (but not exclusively) associated with survival in both archaea and bacteria. Our additional survival experiment (**Fig. 2 & Fig. S3**) further demonstrated low levels of H₂ promotes energy conservation (higher cellular ATP) of cells under starvation-induced persistence.

Fig 3. Many different font sizes make it difficult to really appreciate details. Must zoom considerably to see legend for 3A.

Fig. 3 is a single panel figure without a legend. Therefore, we suspect the reviewer was referring to **Fig. 4A**. To enhance readability of both figures, we have increased the font size for legend for **Fig. 4A** and provided high-resolution vector figures.

General comments:

I would like to have seen H₂ uptake kinetics for *A. brierleyi* grown under anaerobic conditions with S₀ using atmospheric levels of H₂. These are important data to be considered in light of the proteomics results and to better understand the limits of this organism's ability to scavenge H₂.

Atmospheric sources of H₂ will typically be only available in oxic environments where the presence of air provides aeration with oxygen and atmospheric substrates. As such, it is less likely to be significant in anaerobic conditions where atmospheric sources of substrate are inaccessible. However, we agree with the reviewer that the comparison between aerobic and anaerobic uptake will provide insights into the organism's preference and ability to scavenge H₂. We have thus performed an additional experiment characterizing the kinetics of H₂ uptake by *A. brierleyi* under sulfur-dependent organotrophic growth (**Fig. 1D**). The Michaelis-Menten kinetics under anaerobic uptake shows a slightly lower $V_{max(app)}$ (4.27 vs 7.63 mmol g_{protein}⁻¹ h⁻¹) and K_m (0.89 vs 3.67 μM).

We have included the discussion in **L133-135**:

" $V_{max(app),70°C}$ and $K_{m(app),70°C}$ for H₂ uptake are higher during aerobic than sulfur-dependent anaerobic growth (**Fig. 1D**), suggesting this archaeon uses H₂ more quickly under aerobic conditions."

Related to the above comment, I think the title is perhaps too broad. You need stronger data to show that energy is conserved versus a control. Also, you focus only on aerobic trace gas oxidation. So that should be reflected in the title, unless more comprehensive anaerobic data are also included.

We agree and have revised the title more specifically to "**Trace gas oxidation sustains energy needs of a thermophilic archaeon at suboptimal temperatures**". As discussed above, as atmospheric sources of trace gases are only available in oxic environments, we don't think it's necessary to clarify the uptake is aerobic in the title.

Further experiments are defined to test the kinetics of purified proteins, or to perform heterologous expression in other Sulfolobales. I agree these are good suggestions and think it would also be interesting to investigate the effect of mutating these genes and testing the effect on atmospheric H₂ oxidation in whole cells. If this was done, and cells did not grow with trace H₂, it would bolster the argument they are actually conserving energy.

*We respect the reviewer's suggestion that the study of the purified proteins and manipulation of hydrogenase genes will provide detailed mechanistic insights into the controls of atmospheric H₂ oxidation in archaea. However, these experiments are major undertakings beyond the scope of current study and are subject for more focused future studies. At this present time, there are no available tools for the genetic manipulation of *A. brierleyi*; the culture is characterized by low growth yield ($OD_{max} < 0.15$), specific growth requirements (70°C and pH 2), and a preference for low substrate concentrations (growth inhibited at over 0.5 g/l yeast extract). These factors severely limit the scaling of biomass harvesting for protein purification and biochemical characterization. While some Sulfolobales archaea are genetically manipulatable, the heterologous expression of functional [NiFe]-hydrogenases is notoriously difficult due to both the complexity of hydrogenase structural subunits and the requirement of an additional full set of hydrogenase maturation factors (comprising of at least six other genes) to synthesize holo-hydrogenase. As a context, our previous effort to purify and biochemically characterize active Huc hydrogenase in the genetically tractable *Mycobacterium smegmatis* took an 8-year effort (doi.org/10.1038/s41586-023-05781-7).*

*In this revision, we have provided alternative evidence that metabolism of trace H₂ conserves energy for the cells. This is evidenced by significantly higher levels of cellular ATP in cultures supplemented with H₂ including at atmospheric concentrations, especially at 25°C, during the 20-day course of starvation persistence (**Fig. 2D-E; Fig. S3; Table S2**), and the observations of H₂ oxidation mediated by hydrogenases in the zymographic assay (**Fig. 5**). We therefore believe our study has provided sufficient evidence that *A. brierleyi* conserves energy from atmospheric H₂ through the electron transport chain.*

In the methods it says “organic substrate stock solutions. . .” was this really only yeast extract? Or were other organics used that were not specified in the methods? It should also be made clearer this was not added to autotrophic conditions.

Yes, the organic substrate stock solutions refer to yeast extract stock solutions. We have clarified in the revised text and made more explicit that this was not added to autotrophic conditions.

L487-488: “Yeast extract stock solutions (10% w/v) were autoclaved separately before adding to the mineral medium.”

L691-694: “The process was repeated once with the remaining culture. Cells on filter membrane were resuspended and washed by 3 ml of DSMZ medium 150 mineral base (without organic substrates) and the unit was centrifuged for 3 min to remove residual heterotrophic substrates.”

In the materials and methods it mentions that biomass was quantified by OD as well as protein quantification. However, I do not see the protein biomass data presented in the figures in that way. It appears that these data were used for Fig 1C, so perhaps it makes more sense to talk about the protein quantitation in the kinetics section of the methods.

The protein biomass data was used in both H₂ uptake kinetics measurement (Fig. 1C-D) and the new survival assay (Fig. 2D-E, Fig. S3). We feel that it will be more appropriate to describe that we used cell protein to quantify cell biomass in the earlier method section. But we agree that it can be made clearer that protein quantification was used for these two experiments and make the following clarification.

L498-500: *“To quantify biomass of A. brierleyi for H₂ uptake kinetics and survival assays, total cell protein was measured using the bicinchoninic acid protein assay (Sigma-Aldrich) against bovine serum albumin standards.”*

Responses to reviewer #4:

Summary: The genome sequences of the two archaea of interest here clearly indicate the presence of the enzyme types of interest here. While this may have been reported as such in the literature, this comes as no surprise. This report shows the oxidation of low levels of hydrogen and carbon monoxide by the thermoacidophilic archaea *Acidianus brierleyi* and (briefly discussed) *Metallosphaera sedula*. It proposes that the putative genes encoding hydrogenases in *Acidianus brierleyi*, in addition to those already known in the Order Sulfolobales, are responsible for the observed phenomenon. Quantitative proteomics is used to assess the role of these putative hydrogenases in hydrogen oxidation during various modes of growth in *A. brierleyi*. It is pointed out that oxidation of H₂ and CO at trace atmospheric concentrations has only been shown in bacteria previously, and that the archaea also contribute to this cycle.

While this is an interesting observation, it does not ‘move the needle’ much. Yes, these archaea contribute to the cycling of CO and H₂ but at very low levels and mostly in thermal environments. At lower temperatures, the metabolic rates of these archaea are vanishingly small. Since thermal environments make up a very small footprint on earth, the phenomenon described may not matter much in the overall scheme of things.

We thank the reviewer for their comments and consideration of the manuscript. As recognised by the reviewer, one of the key novelties of the current report is the discovery of atmospheric H₂ (and CO) oxidation mediated by members of archaea, overturning the previous paradigm that this metabolism is unique to bacteria. However, we also made significant novel findings regarding this metabolism with major implications on the physiology, evolution, and ecology of archaea. These include:

- 1) The first kinetic characterization showing the constitutive and high-affinity uptake of this gas at nanomolar concentrations including at sub-atmospheric levels;*
- 2) The new finding that these obligate thermophiles maintain atmospheric trace gas oxidation at temperate temperatures far below their growth ranges;*
- 3) New viability assays (added in this revision and elaborated below) showing nanomolar H₂ significantly contributes to the energy conservation and survival of carbon-starved culture with a higher significance at low temperatures (25°C), lending a strong support to;*
- (4) A new energetics-based theory on microbial biogeography that universal atmospheric trace gases promote the survival and thus dispersal of thermophiles across geothermal habitats;*
- (5) The discovery of novel hydrogenases mediating H₂ metabolism in Sulfolobales, including the lineage SUL2 as the primary determinant for aerobic uptake;*

6) *The demonstration of distant relationships between archaeal and bacterial high-affinity hydrogenases, suggesting a divergent origin of atmospheric H₂ oxidation; and*

7) *The substantial evidence on the differential roles of multiple hydrogenases under different redox and growth conditions.*

We agree with the reviewer that the direct implications of these findings for global biogeochemical cycles still need to be clarified. In the revised manuscript, we have cautioned that while these archaea clearly contribute to the cycling of H₂ and CO, this may not be quantitatively significant for the budgets of these gases. However, the observation that trace gas oxidation enables the survival of archaea at a range of temperatures is highly significant, given it suggests a mechanism supporting their global resilience and dispersal.

Comments:

- *Metallosphaera sedula* demonstrates similar trace gas oxidation capabilities to *Acidianus brierleyi*. While *M. sedula* consumes H₂ at a slower rate, it continues to consume H₂ at sub-atmospheric levels while *A. brierleyi* levels off. The authors then show that *M. sedula* and *A. brierleyi* have different types of hydrogenases. In identifying which hydrogenase is primarily responsible for trace gas oxidation in aerobic conditions, it would be helpful to provide proteomic analysis of *M. sedula* for comparison against *A. brierleyi*. Difference in hydrogenase levels/activities could help explain why *A. brierleyi* levels off at low H₂ concentrations while *M. sedula* continues to oxidize H₂.

*The observation that consumption of sub-atmospheric H₂ by *A. brierleyi* levelled off earlier than *M. sedula* more likely reflects different consumption thresholds of high-affinity hydrogenases between the two strains rather than hydrogenase level/activities. This has been consistently observed in atmospheric H₂-oxidizing bacteria showing different H₂ uptake thresholds (e.g. *Mycobacterium smegmatis* ~0.07 ppmv, doi.org/10.1073/pnas.1320586111; *Edaphobacter aggregans* ~0.35 ppmv, doi.org/10.1038/s41396-020-00750-8; *Nitrospira moscoviensis* ~0.12 ppmv, doi.org/10.1038/s41396-022-01265-0). This kinetic property is also evidenced in the current study where *A. brierleyi* H₂ uptake often reached a lower threshold at lower temperatures (higher gas solubility at lower temperatures in solution) (**Fig. S1**). Therefore, it is unlikely that comparative proteomic analysis between *M. sedula* and *A. brierleyi* will provide additional insights in identifying which hydrogenase is primarily responsible for trace gas oxidation in aerobic conditions.*

- The claim that the hydrogenases are 'novel' needs to be supported better. It seems that these fall into the types of these enzymes that have been studied for decades. The 'novelty' may only come into play in terms of thermostability which can be related to the 'non-catalytic' amino acid sequence not involved in the active site.

We discussed throughout the original manuscript that, while the hydrogenases in Sulfolobales are all from the [NiFe]-hydrogenase superfamily, they fall into novel subgroups that are phylogenetically, structurally, and likely functionally divergent from other known enzymes. The H₂-binding active site is likely the same in these enzymes as other [NiFe]-hydrogenases, but the enzymes differ from those previously described in their redox centres and electron acceptor-binding sites. We establish the novelty of the hydrogenases through complementary analyses of sequence divergence, phylogenetic placements, genetic arrangement, and

functional roles in *A. brierleyi*, in light of reports from previous literature. The subgroup classification is conservatively handled and described based on our seminal hydrogenase classification scheme (Greening et al., ISME J 2016; cited over 500 times). We have ensured the revised manuscript makes it clear that Sulfolobales encodes novel previously unidentified [NiFe]-hydrogenase subgroups. We have also revised the manuscript to avoid unnecessarily repeating the term 'novel' when describing the SUL1 and SUL2 subgroups.

As detailed in the original manuscript (Section “**Acidianus brierleyi possesses four phylogenetically and syntenically distinct [NiFe]-hydrogenases widely distributed in Sulfolobales**”), the four [NiFe]-hydrogenases identified in *A. brierleyi* and the wider Sulfolobales archaea, only two were formally described. The group 1g lineage (Hca) was the only experimentally characterized hydrogenase in Sulfolobales while the group 2e (Hys) was defined through genomic surveys in a 2016 report (doi.org/10.1038/srep34212). The other two deep-branching hydrogenase lineages (defined SUL1 and SUL2 in the current manuscript) are previously undescribed and distantly related to reported hydrogenases (Fig. 3). They only share ~30% sequence identity to previously described hydrogenases. SUL2 is also novel and unique in genetic arrangement; while sharing structural genes (*isp1* and *isp2*) similar to *Isp*-hydrogenases (group 1g; group 1e), these two subunits are not found between the large and small subunits (Fig. 4A). We have also provided extensive new insights how these hydrogenases may associate through AlphaFold2 modelling and that they may simultaneously support respiration and carbon fixation through an electron bifurcation mechanism (Fig. 4B-E). Apart from *in silico* analysis, we demonstrated that a key novelty of these hydrogenases is their functional roles: both zymographic experiment and proteomics experiment provide the first evidence that SUL2 serves as the primary hydrogenase responsible for aerobic H₂ uptake; some of these hydrogenases mediate the undiscovered ability to oxidize atmospheric H₂ and this contributes to the thermal tolerance breadth of these archaea. However, their divergence isn't driven primarily by the need to enhance structural stability of these enzymes: rather the entire redox chemistry and physiological integration of these enzymes, especially the dominant SUL2, is completely distinct to any enzyme previously described.

- The authors frequently describe the hydrogenases as “constitutively expressed”, while also showing varying levels of the hydrogenases depending on the mode of growth, implying the hydrogenase expression is highly regulated. These points are contradictory.

We thank the reviewer for the correction and agree that “constitutive” is not the most appropriate word. Our intention was to describe the hydrogenases (SUL2) were constantly produced across various conditions, though its absolute level expression is regulated under a genetic circuit. We have reworded throughout.

- The authors claim that trace gas oxidation could be providing maintenance energy for cells in a low metabolic state. It would be helpful to demonstrate that energy conservation is actually occurring to a biologically relevant extent at atmospheric levels of H₂, either by its effect on cell density or by reduction of an energy carrier through the electron transport chain.

*We thank the reviewer for the excellent suggestion. In this revision, we have provided two lines of additional evidence that metabolism of trace H₂ is used to conserve energy in *A. brierleyi* cells. First, we performed viability assays showing cellular ATP increases with trace H₂ and a zymographic assay showing the SUL2 is the dominant hydrogenase, which oxidizes H₂ and transfers electrons to an artificial electron carrier.*

The viability assay on carbon-starved stationary phase *A. brierleyi* cells supplemented daily with trace H_2 equivalent to 0, 0.6, 5, and 50 ppmv at both 70°C and 25°C for 20 days showed that H_2 supplementation, including at atmospheric concentrations, significantly enhanced cellular ATP, culture density, and cell protein for cultures persisting at 25°C (**Fig. 2D-E; Table S2**). Cultures supplemented with atmospheric H_2 had 38% higher cellular ATP at day 20 than cultures without headspace H_2 . We believe this provides strong evidence that oxidation of the universal trace gases supports energy conservation and potentially the dispersal of aerobic thermophiles through temperate environments. We have now included the new result in **L160 – 179** and expanded the discussion in **L376 – 379** and **L455 – 457**.

Additionally, we performed a hydrogenase activity staining experiment on proteins separated from different cellular fractions of *A. brierleyi* by Blue Native-PAGE gels. Incubated under an atmosphere with H_2 as the only reductant and the artificial dye nitroterazolium blue (NBT) as the only electron acceptor, we consistently detected the H_2 -dependent NBT reduction in bands with a molecular size between 146 – 242 kDa (matching the predicted 211 kDa molecular weight of SUL2 complex) (**Fig. 5A**). Protein mass spectrometry identified SUL2 hydrogenase as the major constituent of the excised gel bands (**Fig. 5B**). The results strongly suggest that SUL2 oxidizes H_2 and transfers electrons to downstream electron carriers for energy conservation. We have now included the new result in **L262-290**. Note that, the test on predicted physiological electron acceptors quinone was precluded due to the insolubility of these highly hydrophobic molecules.

- The structural comparison of the hydrogenases is all computational, making it very difficult to make claims about different types of electron acceptors or the comparative kinetic properties of the hydrogenases. Biochemical characterization of these different hydrogenases from *A. brierleyi* would add substantial support to the authors' claims about the various roles of each hydrogenase. As it stands, the Michaelis-Menten kinetic analysis only evaluates the whole cell behavior and does not provide insight into the differences in substrate affinity between the various hydrogenases.

We understand the considerations by the reviewer. Our structural analysis does not aim to investigate kinetic properties of the hydrogenases, but rather provides important insights on the potential functions and interactions in comparison to known hydrogenases as the foundation for future studies, e.g. the structural similarity of group 2e hydrogenase with the bacterial high-affinity group 2a Huc enzyme, and group 1g and SUL2 hydrogenases with ISP hydrogenases. As discussed above, we conducted additional biochemical characterizations on *A. brierleyi* hydrogenases revealing the dominant H_2 oxidizing activity of SUL2 under aerobic growth conditions, and supporting the AlphaFold2 predicted association of its structural subunits (**new Fig. 5, Fig. S6, Table S4**). The same experiment, however, indicates the growth yield of *A. brierleyi* ($OD_{max} < 0.15$) and the expression of hydrogenases other than SUL2 are key limiting factors for scaling production and purification of these enzymes. Our previous effort to purify and biochemically characterize active Huc hydrogenase in the genetically tractable *Mycobacterium smegmatis* is an 8-year endeavour effort (doi.org/10.1038/s41586-023-05781-7). More elaborate biochemical characterization of *A. brierleyi* hydrogenases is thus beyond the scope of this already extensive study.

Overall, the authors make some insightful claims, but further experimental support would be necessary to solidify the claims about certain hydrogenases being responsible for certain modes of growth or for the trace gas oxidation being able to provide sufficient maintenance

energy to the cells. Also, they should do a calculation that estimates the contribution of these extreme thermophiles to the overall H₂ and CO cycling on this planet.

As outlined above, we provided new experimental evidence that trace gas oxidation provides a significant energy source for carbon-starved cells, which is most relevant at temperatures significantly lower than those required for growth. This is in line with our hypothesis that trace gas oxidation may be an important trait supporting microbial dispersal through temperate environments. Given the multiple uncertainties involved, we do not feel that it is appropriate, or within the scope of this study, to perform scaling calculations of the contribution of these extreme thermophiles to the overall H₂ and CO cycling. We do not claim that extremophilic archaea strongly influence biogeochemical cycles by oxidising trace gases, but rather than trace gas oxidation is a key part of their lifestyle that facilitates survival and likely dispersal. We have ensured that the abstract no longer uses the term “sink” and have added the following caution:

L467-471: *“However, it remains unresolved whether archaeal trace gas oxidisers make a quantitatively significant contribution to the overall biogeochemical cycle of atmospheric H₂ and CO, given they are vastly outnumbered by bacteria in the temperate soils that serve as the main sinks for these gases ¹⁷.”*

Reviewer #1 (Remarks to the Author):

The authors have done a lot of new work, and experiments to address my earlier critical comments. This was appreciated, and I thank the authors for taking my comments into consideration and putting in the extra effort to do the additional experiments. Their new experiments show that the archaea tested can oxidize H₂ aerobically at extremely low H₂ concentrations and also at low temperatures. The new data in the new Figure 2 is convincing and I think goes a long way to support the authors main conclusion, that aerobic oxidation of trace atmospheric H₂ probably helps these archaea to survive dispersal and periods of non-optimal growth conditions. The authors have done a good job tailoring the text to reflect the new data and changes. I therefore recommend the manuscript with the new changes for publication.

Reviewer #2 (Remarks to the Author):

Leung et al have made revisions that address nearly all of my comments. The addition of experiments regarding the influence of trace H₂ on the viability of the target archaeon is particularly helpful. I only have one concern remaining regarding electron bifurcation. Once this relatively small issue is resolved, I have no further comments and endorse publication of this study.

Based on my knowledge, having two electron transfer paths does not imply electron bifurcation. Electron bifurcation involves two "separate" transfers of electrons from a two-electron carrier/donor. The first electron released is low-energy and this leaves a radical intermediate (e.g., Q^{•-} or FAD^{•-}) which is unstable and releases a high-energy electron. This results in two electrons of differing energy levels. Thus, without a two-electron carrier that can form a radical intermediate, one cannot have electron bifurcation (at least based on current knowledge). In the proposed hydrogenase structure, the site where the electron transfer paths diverge is a heme, a single-electron carrier. There are also no two-electron carrier intermediates in the electron transfer path. If electron bifurcation were to take place in this enzyme, the quinol would have to be the donor. However, electron transfer from quinol to hydrogen and ferredoxin are both unfavorable reactions. I believe what the authors potentially have here is a hydrogenase with two alternative electron transfer routes, an alternate interpretation the authors also point out (LN411). Photosynthetic reaction centers are examples of proteins with such schemes. Electron transfer of the heliobacteria reaction center can either donate electrons to menaquinone or ferredoxin. The reaction center of purple bacteria can flow electrons from the first electron-carrying intermediate (bacteriochlorophyll special pair) through one subunit or another depending on the conditions (though the final electron acceptor is the same). Neither of these dividing electron transfer paths are known to involve electron bifurcation.

Minor comments:

LN178: "conserves energy from aerobic oxidation of trace H₂"

LN237, LN404, LN408: the authors repeatedly connect electrons derived from H₂ oxidation to carbon fixation (e.g.), but cytoplasmic reducing power need not feed into carbon fixation – it is also essential for other biosynthetic processes. Phrases like "driving carbon fixation" and "for carbon fixation" are misleading.

LN462: "previously assumed", was there discussion regarding the distribution of atmospheric H₂ oxidation in archaea prior to this study?

- Masaru K. Nobu

Reviewer #3 (Remarks to the Author):

Thank you for the thorough revision through new experimentation to address mine and others' comments. The new data and changes to previous figures better support the arguments made within the manuscript. The additional changes to clarify methods and restrict some previous assertions to be more specific are also appreciated. This is a nice paper that will motivate more

research into archaeal H₂ oxidation.

Reviewer #1 (Remarks to the Author):

The authors have done a lot of new work, and experiments to address my earlier critical comments. This was appreciated, and I thank the authors for taking my comments into consideration and putting in the extra effort to do the additional experiments. Their new experiments show that the archaea tested can oxidize H₂ aerobically at extremely low H₂ concentrations and also at low temperatures. The new data in the new Figure 2 is convincing and I think goes a long way to support the authors main conclusion, that aerobic oxidation of trace atmospheric H₂ probably helps these archaea to survive dispersal and periods of non-optimal growth conditions. The authors have done a good job tailoring the text to reflect the new data and changes. I therefore recommend the manuscript with the new changes for publication.

We are delighted to hear and thank the reviewer for their careful considerations and critical comments that have substantially improved the manuscript.

Reviewer #2 (Remarks to the Author):

Leung et al have made revisions that address nearly all of my comments. The addition of experiments regarding the influence of trace H₂ on the viability of the target archaeon is particularly helpful. I only have one concern remaining regarding electron bifurcation. Once this relatively small issue is resolved, I have no further comments and endorse publication of this study.

We thank the reviewer for his insightful and constructive comments on this manuscript in the previous and current reviews.

Based on my knowledge, having two electron transfer paths does not imply electron bifurcation. Electron bifurcation involves two “separate” transfers of electrons from a two-electron carrier/donor. The first electron released is low-energy and this leaves a radical intermediate (e.g., Q⁻ or FAD⁻) which is unstable and releases a high-energy electron. This results in two electrons of differing energy levels. Thus, without a two-electron carrier that can form a radical intermediate, one cannot have electron bifurcation (at least based on current knowledge). In the proposed hydrogenase structure, the site where the electron transfer paths diverge is a heme, a single-electron carrier. There are also no two-electron carrier intermediates in the electron transfer path. If electron bifurcation were to take place in this enzyme, the quinol would have to be the donor. However, electron transfer from quinol to hydrogen and ferredoxin are both unfavorable reactions. I believe what the authors potentially have here is a hydrogenase with two alternative electron transfer routes, an alternate interpretation the authors also point out (LN411). Photosynthetic reaction centers are examples of proteins with such schemes. Electron transfer of the heliobacteria reaction center can either donate electrons to menaquinone or ferredoxin. The reaction center of purple bacteria can flow electrons from the first electron-carrying intermediate (bacteriochlorophyll special pair) through one subunit or another depending on the conditions (though the final electron acceptor is the same). Neither of these dividing electron transfer paths are known to involve electron bifurcation.

We thank the reviewer for the thoughtful comments on the mechanism of electron bifurcation and the excellent example of heliobacterial photosynthetic reaction centers. Our current models made no assumption that heme is the electron carrier at the diverging site but parts of electron transfer routes. The dual-heme arrangement is reminiscent of the complex III known to mediate quinone-based bifurcation. It cannot be ruled out that a similar mechanism where quinol reduced by H₂ acts as the bifurcation intermediates for both ferredoxins and quinone may occur. In light of current evidence, we agree that the hypothesis that the hydrogenases transfer electrons to different electron acceptors depending on redox conditions is more likely. We have now further nuanced the discussion of electron bifurcation in the main text and expanded the discussion of the alternative hypothesis with the example of heliobacterial RCs (L225-228; L398-405).

Minor comments:

LN178: “conserves energy from aerobic oxidation of trace H₂”

Fixed, thank you.

LN237, LN404, LN408: the authors repeatedly connect electrons derived from H₂ oxidation to carbon fixation (e.g.), but cytoplasmic reducing power need not feed into carbon fixation – it is also essential for other biosynthetic processes. Phrases like “driving carbon fixation” and “for carbon fixation” are misleading.

*We previously limited the discussion on carbon fixation to highlight the role of H₂ in hydrogenotrophic growth of *Acidianus brierleyi*. But we agree with the reviewer that cytosolic reductants are also essential for other biosynthetic processes and have now amended the corresponding sentences to include that cytoplasmic reducing power from H₂ also contributes to biosynthesis (L225, L401).*

LN462: “previously assumed”, was there discussion regarding the distribution of atmospheric H₂ oxidation in archaea prior to this study?

There is no prior discussion of the atmospheric H₂ oxidation in archaea and we agree that “previously assumed” is out of place here. The sentence has now been corrected as “Lastly, atmospheric H₂ oxidation may be a wider metabolic trait among aerobic archaea.”

- Masaru K. Nobu

Reviewer #3 (Remarks to the Author):

Thank you for the thorough revision through new experimentation to address mine and others' comments. The new data and changes to previous figures better support the arguments made within the manuscript. The additional changes to clarify methods and restrict some previous assertions to be more specific are also appreciated. This is a nice paper that will motivate more research into archaeal H₂ oxidation.

We are grateful for the reviewer's time and suggestions for our manuscript, and their recognition of this work in advancing research on archaeal H₂ oxidation.